# A physiological approach for assessing human survivability and liveability to heat in a changing climate

Jennifer Vanos [1] ✉, Gisel Guzman-Echavarria[2], Jane W. Baldwin [3,4], Coen Bongers [5,6], Kristie L. Ebi [7] & Ollie Jay [6]

Most studies projecting human survivability limits to extreme heat with climate change use a 35 °C wet-bulb temperature ($T_w$) threshold without integrating variations in human physiology. This study applies physiological and biophysical principles for young and older adults, in sun or shade, to improve current estimates of survivability and introduce liveability (maximum safe, sustained activity) under current and future climates. Our physiology-based survival limits show a vast underestimation of risks by the 35 °C $T_w$ model in hot-dry conditions. Updated survivability limits correspond to $T_w$~25.8–34.1 °C (young) and ~21.9–33.7 °C (old)−0.9–13.1 °C lower than $T_w$ = 35 °C. For older female adults, estimates are ~7.2–13.1 °C lower than 35 °C in dry conditions. Liveability declines with sun exposure and humidity, yet most dramatically with age (2.5–3.0 METs lower for older adults). Reductions in safe activity for younger and older adults between the present and future indicate a stronger impact from aging than warming.

Adverse health impacts of extreme heat exposure are expected to rise globally due to a warming climate, urban-induced warming, and a growing and aging population[1,2]. The concerns for human health, productivity, and well-being are greater in humid climates and for vulnerable populations[3–5], such as older adults, unhoused, and/or those with chronic diseases. Therefore, robust models to assess current heat-health impacts and project future risks must incorporate specific vulnerabilities and diverse environmental contexts[6].

Methods to project future heat stress risk can be broadly categorized into epidemiology/econometric and physiology-based approaches, which have contrasting benefits and limitations. Epidemiology/econometric approaches are empirical in nature, analyzing time series of historical temperature paired with particular health consequences (e.g., morbidity or mortality) across populations to determine heat-health relationships. These studies often find higher rates of cardiovascular and respiratory deaths associated with high ambient temperatures. Future health burdens from heat can be

estimated by applying these relationships to climate model outputs (i.e., daily temperature) under different warming scenarios[7,8]. Empirical approaches are based on real-life outcomes and the range of realistic living conditions, and they can explore the cumulative effects of exposures over multiple days. However, two limitations for climate change projections include 1) assumptions needed to extrapolate results to warmer temperatures than observed in the historical sample[9] and 2) ambiguity regarding the role of humidity in heat-health outcomes[10]. While some epidemiological studies find a relationship between mortality in the heat and humidity[11], most find minimal associations between humidity and heat-health outcomes[12]. Given that specific humidity is robustly expected to increase with global warming, this uncertainty is a key research gap for epidemiology-based projections of future heat stress.

Physiology-based studies of future heat stress risk employ relationships between the thermal environment and health outcomes based on human energy balance considerations, with parameters

[1]School of Sustainability, Arizona State University, Tempe, AZ, USA. [2]School of Geographical Sciences and Urban Planning, Arizona State University, Tempe, AZ, USA. [3]Department of Earth System Science, University of California Irvine, Irvine, CA, USA. [4]Lamont-Doherty Earth Observatory, Palisades, NY, USA. [5]Department of Medical Sciences, Radboud university medical center, Nijmegen, The Netherlands. [6]Heat and Health Research Incubator, University of Sydney, Sydney, NSW, Australia. [7]Center for Health and the Global Environment, University of Washington, Seattle, WA, USA. ✉e-mail: jvanos@asu.edu

constrained by studies of physiologic processes. In contrast to epidemiology/econometric approaches, physiology-based studies of heat-health outcomes consistently find a robust role of atmospheric humidity in heat stress via its modulation of evaporative cooling from sweat[10]. However, physiology studies are limited in not directly observing health outcomes, such as hospitalization or death, and employ idealized conditions from thermal chamber studies.

A range of physiology-based metrics has been applied to project future heat stress. Sherwood & Huber[12] introduced a 35 °C wet bulb temperature ($T_w$) threshold that would result in death after 6 h of exposure and applied this threshold to project future adaptability limits under varying levels of warming. Since then, numerous studies have used this approach, wherein a psychometric $T_w$ of 35 °C assumes death[13–17]. The $T_w$ of 35 °C represents a thermodynamic limit to heat exchange, whereby the human body becomes an adiabatic system, assuming the person is indoors or shaded, unclothed, completely sedentary, fully heat acclimatized, and of average size without thermoregulatory impairments[12]. As an example of a different metric, Dunne et al.[18] estimated future reductions in labor capacity under different warming scenarios using established guidelines for physical labor under different wet bulb globe temperature levels. While these studies incorporate valuable information about humidity and physiology more realistically than epidemiological studies, their thermal physiology theory remains relatively unsophisticated. These approaches cannot capture complexities and personal characteristics affecting human thermoregulation (e.g., body size, activity levels, clothing, or physiological restrictions—such as sweating—to thermoregulation[6,19]), which may cause substantial errors.

To be useful, heat-health projections should realistically account for factors that increase health risks, such as individual physical characteristics and physiological impairments, as well as interventions that modify or decrease impacts (e.g., lowering metabolic rate; behaviors to reduce exposures). Moreover, models should incorporate ranges in environmental parameters that, together with temperature, result in specific thermoregulatory effects (e.g., dry, humid, sun/shade, windspeed)[19]. Physiological and biophysical models offer new opportunities to assess how humans might live and work in a warmer future rather than merely determining the prospects for life and death. Here, we demonstrate a unique approach using physiological principles that align with human thermal responses to heat (e.g., heat strain) to overcome simplified approaches that miss essential physiologic and behavioral factors in the heat.

Heat strain, characterized by thermal, cardiovascular, and renal strain, can lead to adverse health outcomes such as heat exhaustion, heat stroke, or cardiovascular collapse. Risk is higher in people with pre-existing illnesses (e.g., cardiovascular disease[20], or are immunocompromised[21]). Aging[1,21], certain medications[22], and differences in body composition[23] may exacerbate heat strain due to impairments to sweating and/or blood flow or the internal management of body heat storage. Conversely, fitness[24], heat acclimatization[25], and behavioral adaptations[26–28] protect against excessive heat strain. This study and others assessing survivability limits (e.g., refs. 12,14,16) estimate deaths from critical high core temperatures causing heat stroke, where a complex cascade of events[29] ultimately leads to multi-organ failure[30,31] and often death[29]. Hence, we model heat stroke deaths (hyperthermia) and do not model the two other common types of heat-related deaths: cardiovascular collapse and renal failure, acknowledging that heat stroke deaths are a fraction of total excess heat-related deaths.

A human would experience heat stroke death from hyperthermia on 99.9% of occasions when an individual's core temperature ($T_{core}$) exceeds 43 °C[29,32] (see Table 1). Thus, we define the limit of survivability as reaching a $T_{core}$ of 43 °C in 3- or 6-hour exposure windows to allow for comparison with the $T_w$ of 35 °C assumption (heat stroke death after 6 hours). Liveability is the maximum internal heat production, or level of physical activity, that a person can generate without a sustained positive rate of heat storage in the prevailing environment, thus allowing safe, sustained work and play for an extended period. The realistic final $T_w$ value for the limits of survivability or liveability will differ by person (age, body size) and climate (dry versus humid; sun versus shade), and thus is flexible (i.e., the limits will differ). Hence, while we state a final $T_{core}$ at which heat stroke death will almost inevitably occur, our approach does not assume a unique $T_w$ threshold; rather, hundreds of $T_w$ thresholds are possible depending on differences in people and conditions modeled, with wide-ranging opportunities of the model in future research.

The overarching goal of this study is to improve heat survivability modeling frameworks and approaches used in climate change research and introduce an approach to assess liveability. As an initial step towards these advances, we apply a whole-body human heat exchange model to estimate heat stroke deaths and maximum/safe activity intensity. We focus on two subpopulation types (younger (~18–30 y) and older (>65 y) female adults), considering sweat rate impairments (due to older age) and heat exposures in the sun or shade across an array of air temperature ($T_{air}$) and humidity levels. We estimate 1) a range of survivability conditions for humans, particularly where deviations from the 35 °C $T_w$ model exist, and 2) the liveability of humans based on levels of physical activity that can be safely carried out without sustained positive heat storage. Finally, we apply the new liveability approach using global climate model (GCM) data to estimate liveability at a global scale, under current and projected climates.

An overview of environmental conditions, populations modeled, and assessment types included herein is provided in Fig. 1. We do not demonstrate the sensitivity to illness or health status (including acclimatization) in the population, activity velocity, personal cooling strategies, or an ensemble of climate projections—investigations that are worthy future extensions of the methods described in this paper, but out of scope for the present work (see below).

## Results

### Survivability

All survivability results are presented by age (younger versus older female adults) for discrete 3- and 6-hour heat exposure durations. These durations are consistent, and thus comparable, with past studies assessing future survivability[12] and the highest time resolution data commonly saved from GCM simulations. Results are shown for moderate to extreme combinations of temperature and humidity that may occasionally be reached today, both indoors and outdoors, but more frequently in the future[32]. Further, we acknowledge that in most circumstances, human agency allows people to seek shade to reduce their heat load[33]; hence, we primarily focus on shaded survivability results, allowing a more direct comparison with the $T_w$ of 35 °C adaptability hypothesis.

The updated physiologic survivability curves—or the environmental limits indicating heat stroke—by exposure duration and age group are shown in Fig. 2 (thick black lines). These environmental ranges, or zones, indicate that survivability are larger for young adults (Fig. 2a). As conditions become drier, the survivability area declines and the curves bend down due to sweating restrictions, a feature not captured by the traditional $T_w$ of 35 °C approach. These shifts are observed at the intersection between zones 3 (pink), 4 (purple), and 5 (gray) in Fig. 2, each indicating a different reason for the increase in $T_{core}$ towards death—i.e., due to environmental restrictions ($E_{max_{env}}$) (pink), sweating restrictions ($E_{max_{sweat}}$) (purple), or both ($E_{max_{env}}$ and $E_{max_{sweat}}$) (gray) (see Methods and SM).

The 6-hour survival limit for $T_w$ differs depending on the magnitudes of $T_{air}$ and relative humidity (RH). For example, at an RH of 50%, a healthy young adult can withstand conditions up to 43.3 °C $T_{air}$ (representing a $T_w$ of 33.6 °C). However, under drier conditions (RH of 25%), the same young, healthy adult may withstand a $T_{air}$ up to 49.9 °C (or a $T_w$ of only 31.3 °C). The survivability in these dry conditions is

**Table 1 | Different definitions and approaches to survivability and liveability (including terminologies) in recent climate change and physiology literature**

| Term | Definition/Use | References |
|---|---|---|
| Survivability | Individual's core temperature exceeds 43 °C within a 3- or 6-h exposure window. | Current paper |
| Adaptability[a] Limit | An extreme upper limit of $T_w$ = 35 °C to human heat adaptation due to climate warming, above which dissipation of heat becomes impossible when exposed for extended periods (6 h), resulting in death. | Sherwood and Huber[12] |
| Liveability[a] | The maximum safe internal heat production/physical activity that a person can generate without a sustained rate of positive heat storage in the prevailing environment, thus allowing people to safely sustain work or play for an extended period. | Current paper |
| Critical $T_w$ | Critical environmental limits ("adaptability threshold") for compensability (the definition of liveability in current paper), valid for 1.8 METs (light, everyday activity). | Vecellio et al.[39] |

[a]The term habitability may be used as a way to refer to liveability or adaptability in some studies as well.

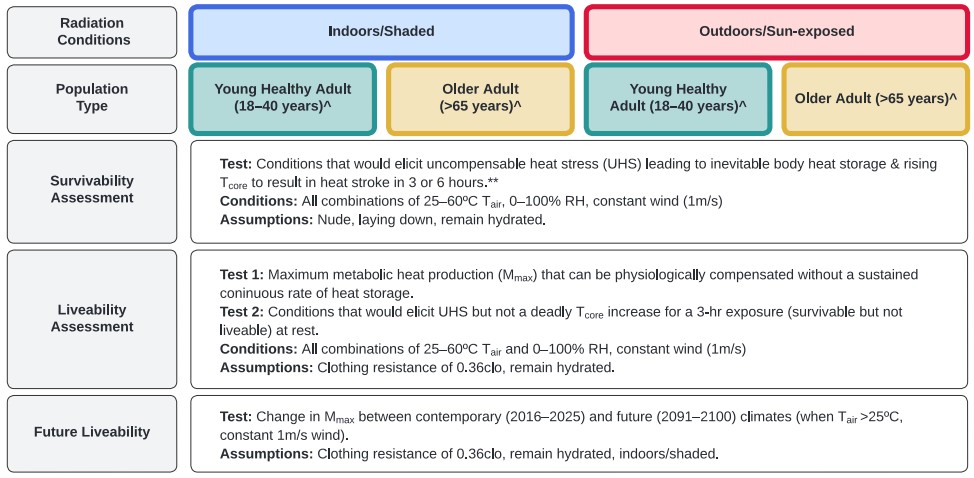

^ Assumptions on population characteristcs for young and old female adults, respectively: average female body sizes by age (1.60m² & 1.78m²), constant skin wettedness (0.65 & 0.85), and maximum sweat rates (0.51 & 0.75L/hr).

** Uncompensable heat stress with limited heat storage.

**Fig. 1 | Overview of environmental conditions, populations, assumptions, and assessment types examined in the present study.** Full model presented in Supplementary Information. All variables in the model remain constant during the exposure (e.g., wind, clothing, sweat rate, skin wettedness). $T_{core}$ Core temperature, $T_{air}$ Air temperature, RH Relative humidity.

impacted by the body's ability to produce sweat, as visualized by the growth of zone 4 (curve shifting down) in Fig. 2 as conditions become drier. For older adults, this downward shift is prominent at a lower $T_{air}$ and higher RH, where these older individuals can survive only a 6-hour exposure at a $T_{air}$ of 45.4 °C at 25% RH ($T_w$ of 27.8 °C), which is 3.5 °C $T_w$ lower than for young adults, and 7.2 °C lower than the 35 °C $T_w$.

When expressing the survivability limits of the physiological model in terms of the critical $T_w$ within very humid (RH > 90%) and shaded conditions, values are broadly in agreement with the traditional $T_w$ = 35 °C assumption ( ~ 0.7–1.3 °C lower $T_w$ for a 6-hour duration) (see Table 2, Fig. 3). However, in drier conditions, the physiological survival $T_w$ limits are much lower (distance between survivability line and $T_w$ = 35 °C line, Fig. 3). As such, the most critical differences from the $T_w$ = 35 °C assumption occur as conditions become very hot and dry ($T_{air}$ > 40 °C and RH < 25%). For young adults, the physiological $T_w$ survival limit ranges from 25.8–31.3 °C (or 3.8–9.2 °C lower than 35 °C). These dry-condition limits are reduced further for older adults to a $T_w$ of 21.9–27.8 °C (or 7.2–13.1 °C lower than 35 °C). Further, shaded older adults would not survive a 6-hour exposure beyond $T_w$ = 21.9 °C at 10% RH (thus $T_{air}$ = 46.4 °C); yet the $T_w$ = 35 °C assumption presumes they could endure a 6-hour exposure at a $T_{air}$ of 60 °C+ at the same RH.

**Liveability**
The limit to safe physical activities differs by $T_{air}$, RH, age, and sun exposure (Fig. 4). $M_{max}$, represented by metabolic equivalents (METs),

is higher under cooler, drier conditions. In shade at $T_{air}$ = 25 °C, young adults can safely reach $M_{max}$ values around 5.0 METs (e.g., dancing) for humid conditions while maintaining thermal equilibrium, with even higher possible $M_{max}$ (8.4 METs, or running) in drier environments. When sun-exposed at the same $T_{air}$, the $M_{max}$ limit decreases to ~3.9–7.4 METs (e.g., housework, climbing stairs). As $T_{air}$ rises in Fig. 4, the $M_{max}$ declines sharply for RH above 75%. In such humid conditions, young adults reach a limit in their ability to perform any activity safely at a $T_{air}$ of ~35.5 °C (or 34.0 °C $T_{air}$ for older adults). Accordingly, the hatched zone in Fig. 4 highlights the range of conditions in which no additional work >1.5 METs can be safely performed. The lower limit corresponds to the transition from compensable, or safe heat stress, to uncompensable heat stress when the internal body temperature starts to rise.

$M_{max}$ decreases with age (Fig. 5), a factor that is considerably higher under dry conditions. For example, at high $T_{air}$ and low RH, young and healthy adults can do ~2.5–3.0 METs more activity (e.g., sitting/resting versus walking) than older adults, who are more limited by their sweat rate. The differences between the two groups decrease (down to ~0.6–0.8 METs less) as humidity rises and temperatures drop. Accordingly, there is a wider range of conditions where older people can only rest safely (i.e., survivable but not livable). Direct solar radiation exposure also decreases $M_{max}$ to a certain extent in both groups (Figs. 4c, d and 5b). This effect is increasingly worse in more humid environments for any $T_{air}$. As a reference, under the same $T_{air}$ and RH, sunlit exposures reduce activity levels by 1.0 MET in young adults and 0.86 METs in older adults compared to shaded.

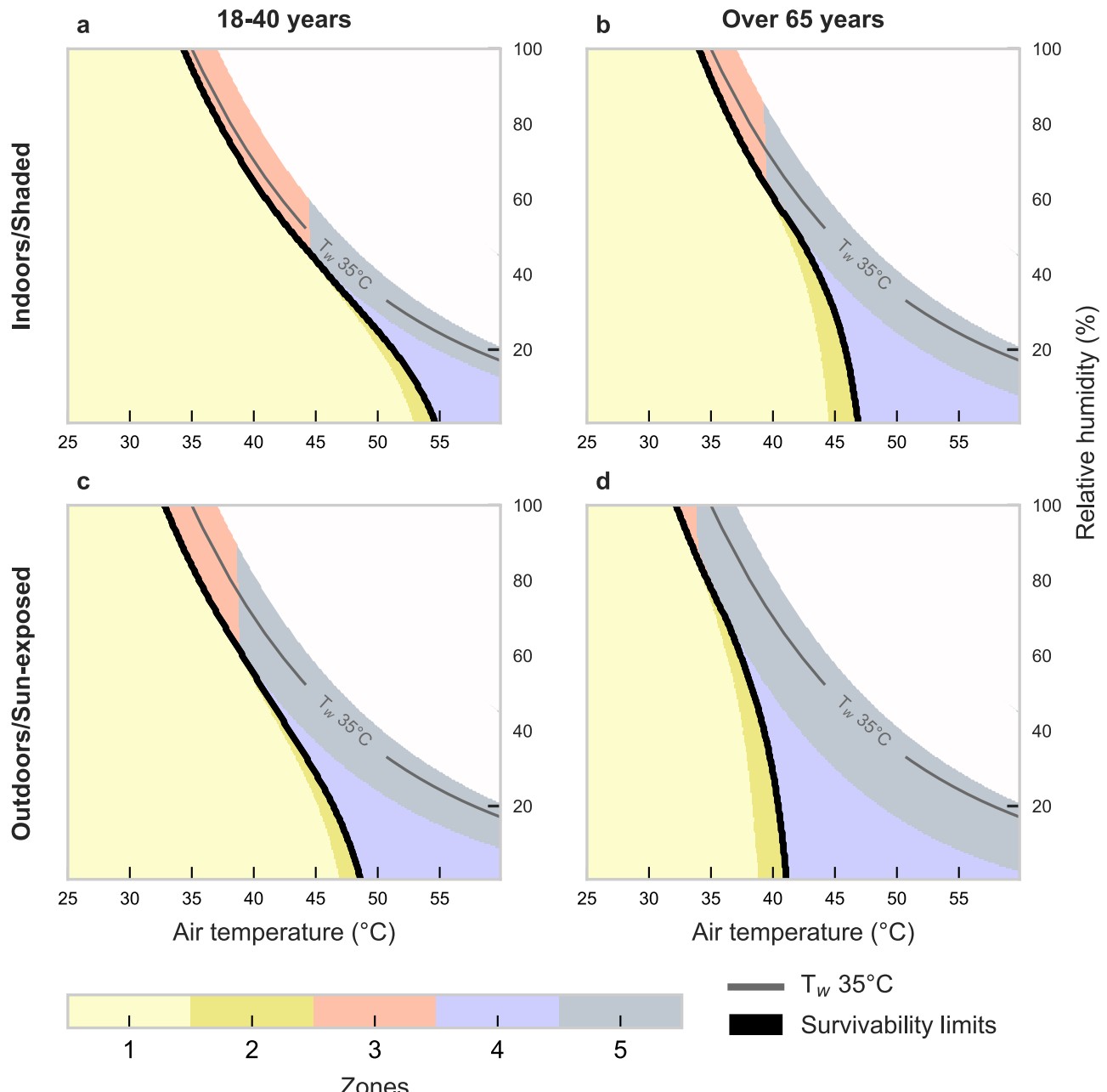

**Fig. 2 | New T$_w$ survivability limits for younger and older adults under shaded/indoor or sun-exposed conditions.** Physiological heat stroke T$_w$ survivability limits (thick black line), modeled across 6 h of constant exposure for young adults (**a, c**) and older adults (**b, d**) in shaded/indoor (**a, b**) and sun-exposed/outdoor (**c, d**) conditions. Zones 1 and 2 (yellow shades) represent areas of survivability, whereas zones 3–5 are non-survivable areas due to evaporative restrictions from the environment (zone 3), sweating limits (zone 4), or both (zone 5) (see also Supplementary Fig. S1). T$_w$ = 35 °C line is shown by the thin gray line. These physiological survivability limits illustrate the environmental conditions in which the body would reach a deadly T$_{core}$ (i.e., after reaching uncompensable heat stress and accumulating enough heat to increase base T$_{core}$ by 6 °C, from 37 °C (normothermia) to 43 °C (heat stroke)). (See Supplementary Figs. S2 and 3 for vapor pressure and specific humidity on y-axis, respectively). Source data are provided as a Source Data file.

## Global climate projections using liveability analysis

Results are presented here for GFDL ESM4; all MPI ESM1.2 results can be found in the SM. Significant declines in $M_{max}$ between present-day (2016–2025) and end-of-century (2091–2100) are projected (Fig. 6), with exceptions in some locations above 50°N, the Himalayas, south New Zealand, and the Andean Cordillera. These global estimates suggest a likely average (median) decrease in $M_{max}$ (from present-day to end-of-century) of −0.25 METs following SSP2-4.5 for young, healthy adults (Fig. 6a) during days when T$_{air}$ > 25 °C (−0.64 METs for SSP5-8.5).

The median $M_{max}$ for the current decade when T$_{air}$ is >25 °C (Fig. S.9) is higher (up to 5.9 METs) across extratropical and mountainous regions (e.g., Ethiopia and the Andean Cordillera in northern South America). In contrast, the lowest $M_{max}$ estimates, reaching an average minimum of 4 METs, are found across coastal areas and the most humid regions in the tropics (i.e., northern India and Bangladesh, the Amazon and Congo Rainforests, southeastern Asia, Eastern China, Gulf of Mexico). In such areas (and other regions covering the lower quartile with major $M_{max}$ changes), $M_{max}$ average declines of >0.3 METs following SSP2-4.5 are expected by 2100 (see Fig. 6a, b) for

**Table 2 | Physiological limit based on our physiological wet-bulb survival temperature ($T_w$), as modeled in the present study, and differences from the 35 °C $T_w$ survivability assumption [i.e., ($\Delta T_w = T_w{-}35$)]**

| Shaded/Indoors | | | | | | | | |
|---|---|---|---|---|---|---|---|---|
| **Exposure time** | **Age** | **Relative humidity** | **10%** | **25%** | **50%** | **75%** | **90%** | **100%** |
| 3-hour | Young 18–40 yr | $T_w$ limit to survive (°C) | 26.7 | 31.9 | 34.3 | 34.7 | 34.8 | 34.9 |
| | | $\Delta T_w$ ($T_w{-}35$) (°C) | (−8.3) | (−3.1) | (−0.7) | (−0.3) | (−0.2) | (−0.1) |
| | | Corresponding $T_{air}$ (°C) | 54.7 | 50.7 | 44.0 | 38.7 | 36.3 | 34.9 |
| | Older >65 yr | $T_w$ limit to survive (°C) | 23.14 | 29.19 | 33.2 | 34.27 | 34.62 | 34.8 |
| | | $\Delta T_w$ ($T_w{-}35$) (°C) | (−11.9) | (−5.8) | (−1.8) | (−0.7) | (−0.4) | (−0.2) |
| | | Corresponding $T_{air}$ (°C) | 48.6 | 47.2 | 42.8 | 38.3 | 36.1 | 34.8 |
| 6-hour | Young 18–40 yr | $T_w$ limit to survive (°C) | 25.8 | 31.3 | 33.6 | 34.0 | 34.1 | 34.3 |
| | | $\Delta T_w$ ($T_w{-}35$) (°C) | (−9.2) | (−3.8) | (−1.4) | (−1.0) | (−0.9) | (−0.7) |
| | | Corresponding $T_{air}$ (°C) | 53.2 | 49.9 | 43.3 | 38 | 35.6 | 34.3 |
| | Older >65 yr | $T_w$ limit to survive (°C) | 21.9 | 27.8 | 32.6 | 33.4 | 33.7 | 34.0 |
| | | $\Delta T_w$ ($T_w{-}35$) (°C) | (−13.1) | (−7.2) | (−2.4) | (−1.6) | (−1.3) | (−1.0) |
| | | Corresponding $T_{air}$ (°C) | 46.4 | 45.4 | 42.1 | 37.4 | 35.2 | 34.0 |

Data are shown for varying relative humidity levels under shaded conditions, along with corresponding air temperature. Each humidity level is shown by the dark blue horizontal lines on Fig. 3. Differences are stratified by exposure duration (3 or 6 h) and age group, depicting a vast underestimation of impacts using traditional $T_w$ of 35 °C limit. Values for sun-exposed/outdoor conditions are in Supplementary Table S.1.

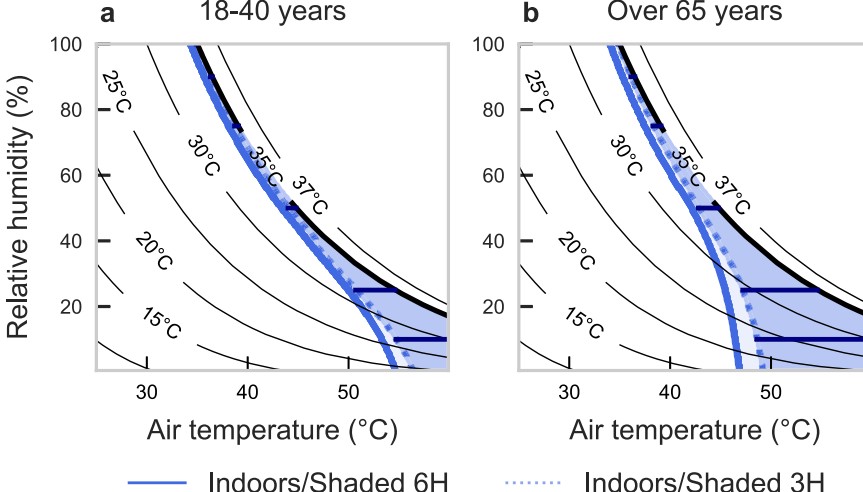

**Fig. 3 | Updated survival limits (thick blue line) in extreme heat versus the $T_w$ of 35 °C assumption (black thick line).** Comparison of the $T_w = 35$ °C assumption (black) with our physiological survivability limits (blue), based on 3-hour (dashed lines) and 6-hour (solid lines) of constant exposure at given air temperature and relative humidity combinations within shaded/indoors conditions. Graphs indicate limits to survivability for (**a**) young and (**b**) older adults, wherein the limit is based on reaching uncompensable heat stress and accumulating enough body heat for a fixed time of constant exposure to increase $T_{core}$ by 6 °C (from 37 °C (normothermia) to 43 °C (heat stroke)). Black lines show $T_w$ values up to 37 °C to avoid unrealistic conditions (thick line shows $T_w$ of 35 °C), and horizontal dark blue lines indicate the $\Delta T_w$ values at RH levels of 10, 25, 50, 75, and 90, and 100% in Table 2. See Supplementary Figs. S4 and 5 for vapor pressure and specific humidity on y-axis, respectively. Source data are provided as a Source Data file.

young adults, which would more than double under SSP5-8.5 (−0.73 METs).

Following SSP2-4.5, large areas will experience >5% reduction in median $M_{max}$ towards the end-of-century (Supplementary Fig. S.11) during warm conditions, including warm coastal areas and savannahs in North, South, and Central America, the Caribbean, the Sahel, much of Eastern Africa, the Arabian Peninsula, Southeast Asia, northern Australia, the Amazon, southeastern U.S., and scattered areas in Central and Eastern Europe, and Central and Eastern Asia. Many of these areas already experience frequent $T_{air} > 25$ °C (>60% of the time) and already-low $M_{max}$ (Fig. 6c, Supplementary Fig. S.12); the further reduction in $M_{max}$ will add major liveability challenges as climate change progresses. Finally, by end-of-century, select areas (Arabian Peninsula, Northern India, Bangladesh) are projected to see a

significant increase in conditions that are survivable but not livable for young adults, reaching a frequency of 5–7% (6 months/decade) for SSP5-8.5 (see hatched zones in Fig. 4, Supplementary Information (SI) S.13).

The $M_{max}$ distribution for select locations in Fig. 7 shows a more pronounced decrease in safe activity in older versus younger adults compared to the decline expected from projected warming (i.e., individual age and age of the overall population can be a stronger heat risk predictor than warming due to climate change). For regions around Riyadh, Cartagena, New Delhi, and Dhaka, there is already a high frequency of time where an older adult cannot presently perform more than 2.5 METs of activity (slow walking). Thus, moving forward, these locations will increasingly become unliveable despite being survivable (increases of frequencies at 1.5 METs or less for old adults,

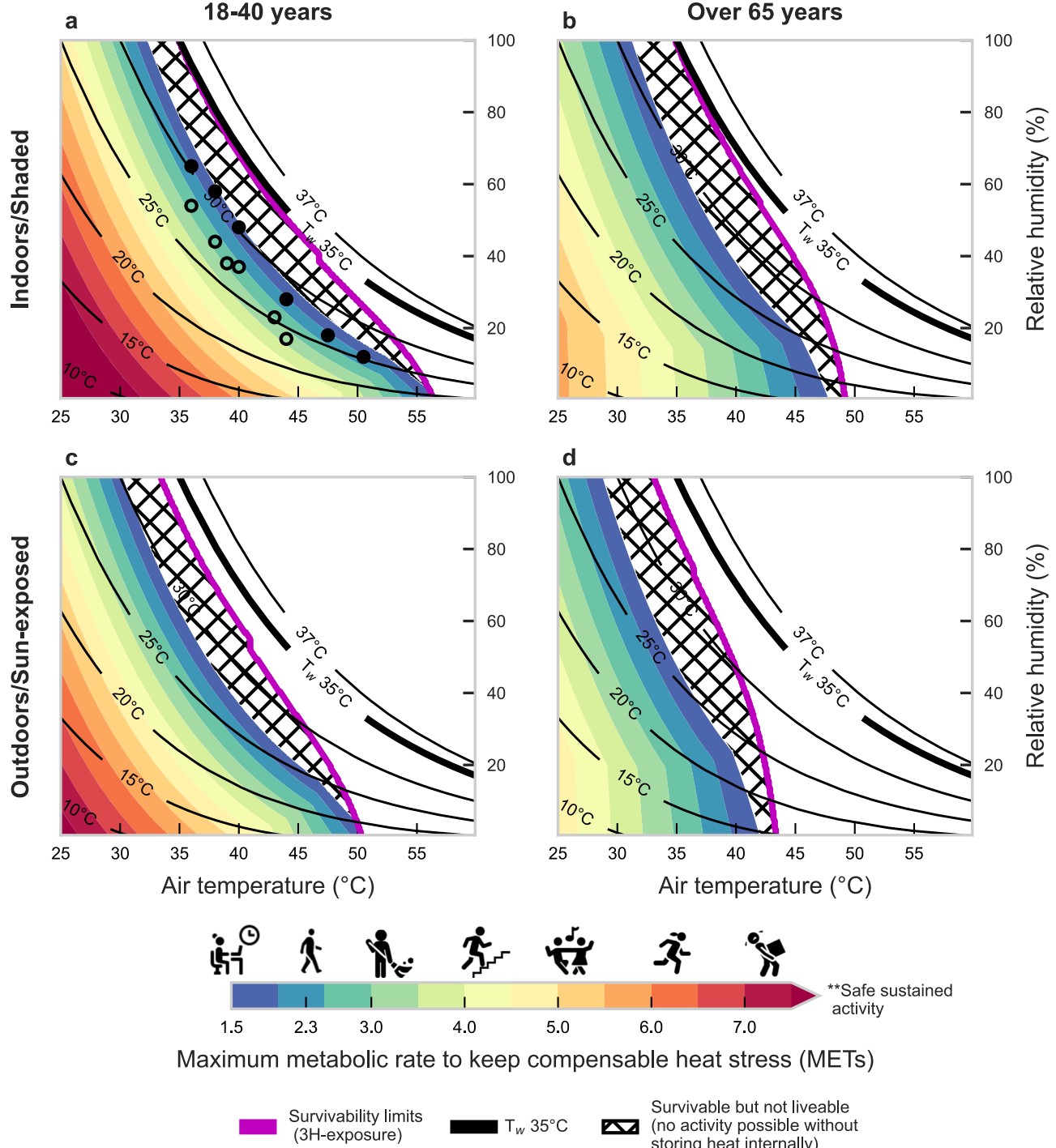

**Fig. 4 | Estimates of liveability at varying combinations of air temperature and relative humidity.** Liveability estimates based on maximum safe metabolic rate ($M_{max}$) that a person can generate without a sustained positive rate of heat storage even with a maximal thermoregulatory response. Results are presented across a range of air temperature and relative humidity for younger (**a**, **c**) and older adults (**b**, **d**) in shaded (top) or sun-exposed (bottom) steady-state environments. The new 3-hour survivability line is shown in purple; constant $T_w$ values are shown by the solid black lines until 37 °C to avoid unrealistic conditions, with $T_w = 35$ °C shown by the thick black line. Activities by MET level range from no activity (sitting -1.5 METs), to housework (-3.0 METs), dancing (-5.0 METs), and heavy lifting (-7.0 METs). The hatched area indicates conditions that are survivable but not livable (i.e., people cannot increase their activity without continuously storing heat inside the body, which will lead to a continuous rise in core temperature, but heat stroke death after a 3-hour exposure would not occur). Icons indicate MET-equivalent activities according to Ainsworth et al.[56]. Circles indicate critical $T_w$ limits reported by Wolf et al.[62] for minimal (-1.8 METs–filled circles) and light physical activity (-3.2 METs–open circles). Note: 1 MET corresponds to complete rest. See Supplementary Figs. S6 and 7 for vapor pressure and specific humidity on y-axis, respectively. Icons provided by Icons8 (https://icons8.com). Source data are provided as a Source Data file.

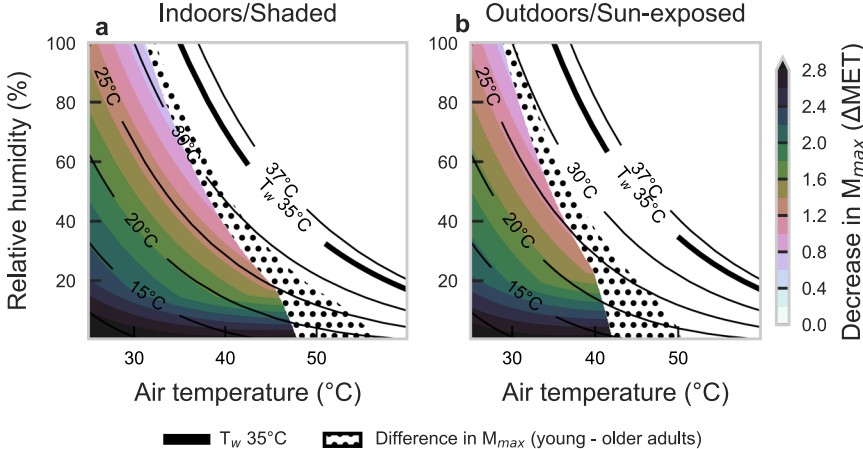

**Fig. 5 | Difference in liveability between older and younger adults for shaded or sun-exposed conditions.** The difference in liveability−or maximum safe metabolic rate ($M_{max}$)−between young and older adults in sun-exposed (**a**) and shaded/indoor (**b**) conditions. The dotted area indicates air temperature and relative humidity combinations that, with aging, shift the situation from liveable to only survivable (i.e., people cannot increase their activity without continuously storing heat inside the body, which will lead to a continuous in core temperature, but heat stroke death after a 3-hour exposure would not occur). For example, in the shade, a young adult can perform 1 METs more work that an older adult within the green-shaded area. Note: The ΔMET values can also be seen by directly comparing Fig. 4a, b (for 5a) or Fig. 4c, d (for 5b). Source data are provided as a Source Data file.

especially following SSP5-8.5. See also Supplementary Figs. S9–S15 for further context.

## Discussion

We report a new modeling approach for assessing the prospects of human survivability and liveability due to extreme heat exposure that can be applied in any climate regime and customized with population groups with potential co-morbidities or thermoregulatory impairments. This new approach integrates well-established and fundamental principles from thermal physiology and human biophysics and accommodates 3- and 6-hour exposure windows aligning with outputs from climate models and past survivability studies. Results encompass current and future extreme heat (see Supplementary Figs. 14 and 15 for future liveability ranges) across very hot and dry and very hot and humid conditions, with risks increasing, or expected to increase, over most of the world[32,34].

For the past decade, the singular psychometric $T_w$ threshold of 35 °C (e.g., refs. 12,16,35) has been the conventional method for assessing the survival limits of humans exposed to extreme heat with climate change. Under such conditions, dry and evaporative heat transfer avenues are abolished. Unable to dissipate any heat, the body would retain all internally generated metabolic heat, inevitably leading to heat stroke death within a 6-hour timeframe. While this approach incorporates biophysical principles, it omits human thermal physiology aspects (e.g., sweat response, hydration, acclimatization), and cannot capture complexities of thermoregulation (e.g., body size, activity, or physiological restrictions[6,19]), which the proposed physiological model begins to overcome.

Omitting thermoregulatory responses to extreme heat can result in a vast overestimation of the limits of human heat tolerance (Fig. 2). Here, we show that under very humid conditions, the difference between models−expressed as a difference in the critical $T_w$ at which humans are projected to survive−is modest (~1 °C). Indeed, Sherwood and Huber[12] and others[36] acknowledge the danger of moist heat. However, many users of the 35 °C $T_w$ limit (e.g., refs. 15,35) have recognized that it defines a threshold for human survivability or adaptability for the best-case scenario (i.e., highly fit, nude, well-ventilated, and shaded conditions)[37]. Thus, the $T_w$ threshold is likely lower for most people, which agrees with recent work by Vecellio et al.[38] At higher $T_{air}$ accompanied by lower humidity, drastic differences (e.g., $\Delta T_w$ ~ 4–13 °C) between the $T_w$

35 °C assumption and our survivability approach emerge (Fig. 2a), similar to climate chamber findings by Vecellio et al.[39] who applied their empirical findings with global climate models[38]. Our model accounts for realistic sweat production (and therefore evaporative potential) to physiologically plausible limits[40–42], which is the underlying reason for the increasing dissociation from the $T_w$ 35 °C limit in hot, dry air. The $T_w$ 35 °C threshold assumes no possible sweat evaporation over the skin and thus theoretically obviates the effect of any potential differences in sweating capacity. However, impossibly high sweat rates are needed for survivability in very hot and dry conditions, resulting in $T_w$ survivability values considerably lower than 35 °C. Higher wind speed can also affect evaporative cooling when sweat rates are sufficient.

Differences between the two approaches widen when estimating survival limits for older adults (Figs. 2 and 3), especially in hot and dry conditions, due to age-related impairments in sweating that are routinely observed above the ages of ~60–65 years[43,44]. Given that we discount potential co-morbidities that may further hinder thermoregulation, the actual upper sweating and evaporative heat loss limits may be even lower than estimated here.

Additional solar load (Fig. 2c, d) also decreases survivability, which is not accounted for in the 35 °C $T_w$ model, chamber $T_w$ threshold studies[39], and many other common bioclimate indices. In a survival scenario, most people would likely behaviorally adapt by seeking the coolest available place, which would inevitably be shaded[45]. There are plausible exceptions to this assumed capacity to avoid extended sun exposure: those without access to shelter or those unable to respond adequately due to health conditions impairing decision-making and/or a lack of mobility. Leading risk factors for heat-related mortality and morbidity in present-day heatwaves include being unhoused or experiencing homelessness[46,47], physical disability[48], and mental health illnesses or behavioral disorders[49–51]. Nevertheless, the outputs for shaded conditions are the most reasonable survivability comparison between our model and the 35 °C $T_w$ model (Fig. 2a, b). The extended heat index (based on the original equations by Steadman's apparent temperature approach[52]) also presents an alternative method to the $T_w$ of 35 °C approach for assessing heat stress[53].

The 35 °C $T_w$ survivability limit assumes heat stroke death after a 6-hour exposure[12]. Depending on the resting metabolic rate and assuming a normothermic resting $T_{core}$ of ~37 °C, after 6 hours of

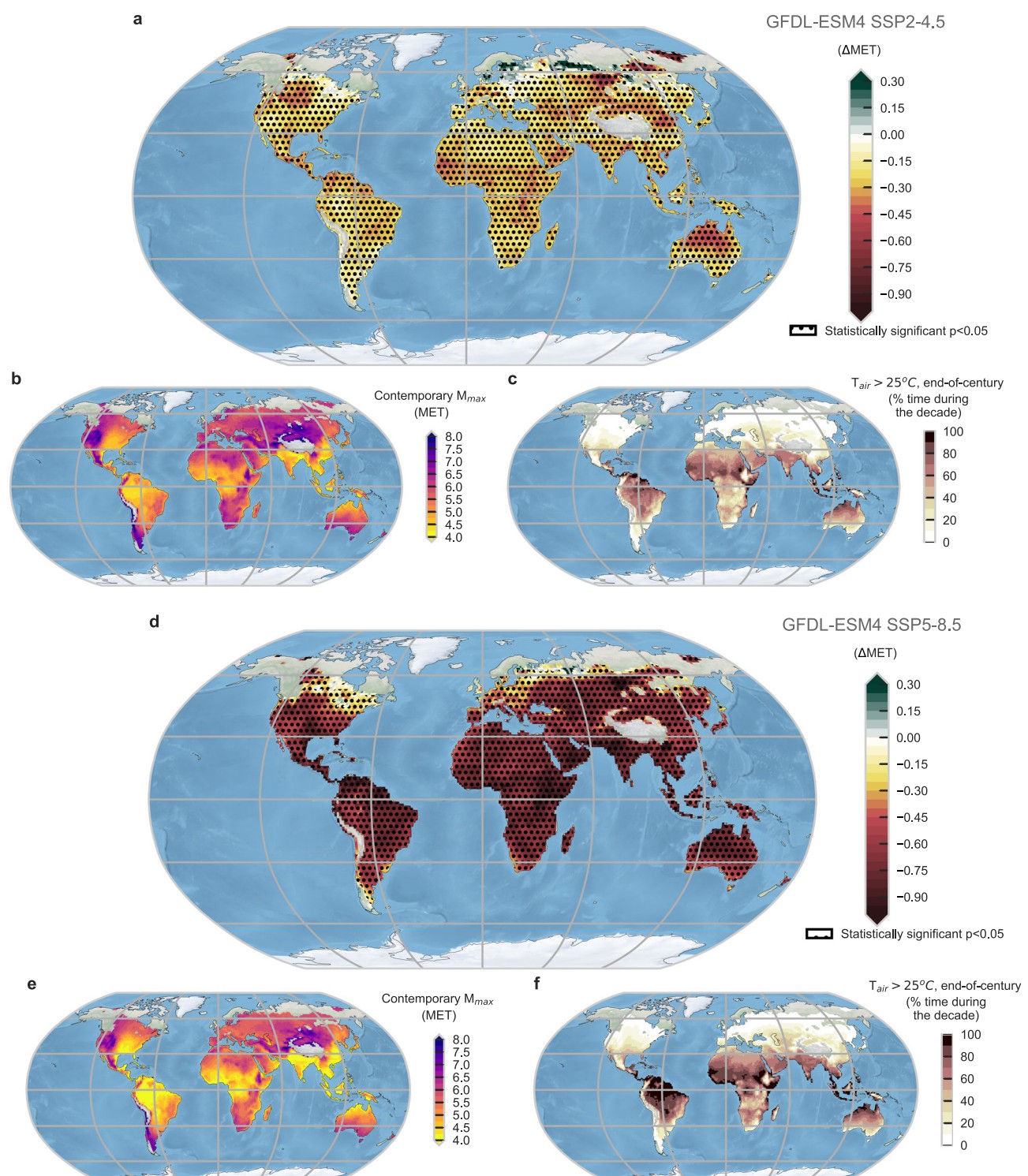

**Fig. 6 | Global maps of liveability estimates. a, d** Differences in median safe $M_{max}$ (maximum safe metabolic rate) (ΔMET) between the present (2016–2026) and end-of-century (2091–2100) for SSP2-4.5 and SSP5-8.5, respectively (negative values indicate less activity possible), with dotted areas indicating locations where median difference is significantly different ($P < 0.05$), (**b, e**) Median $M_{max}$ for current decade (2016–2026) for SSP2-4.5 and SSP5-8.5, respectively, where lower values (e.g., in Bangladesh) indicate the most oppressive conditions and least ability to perform activity, (**c, f**) the percentage of time for 2091–2100 decade with $T_{air} > 25\,°C$ for

SSP2-4.5 and SSP5-8.5, respectively. All analyses are based on warm conditions ($T_{air} > 25\,°C$) for young adults. 3-hourly CMIP6 data are from GFDL ESM4 (-1° × 1.25° atmosphere/land grid) following SSP2-4.5. Areas with no data indicate locations that do not reach $T_{air} > 25\,°C$ in the given decade. Note: 1 MET corresponds to complete rest. CMIP6 Coupled Model Intercomparison Project phase 6, GFDL ESM4 Geophysical Fluid Dynamics Laboratory Earth Systems Model 4. SSP Shared socioeconomic pathway. Made with Natural Earth - free vector and raster map data at naturalearthdata.com. Source data are provided as a Source Data file.

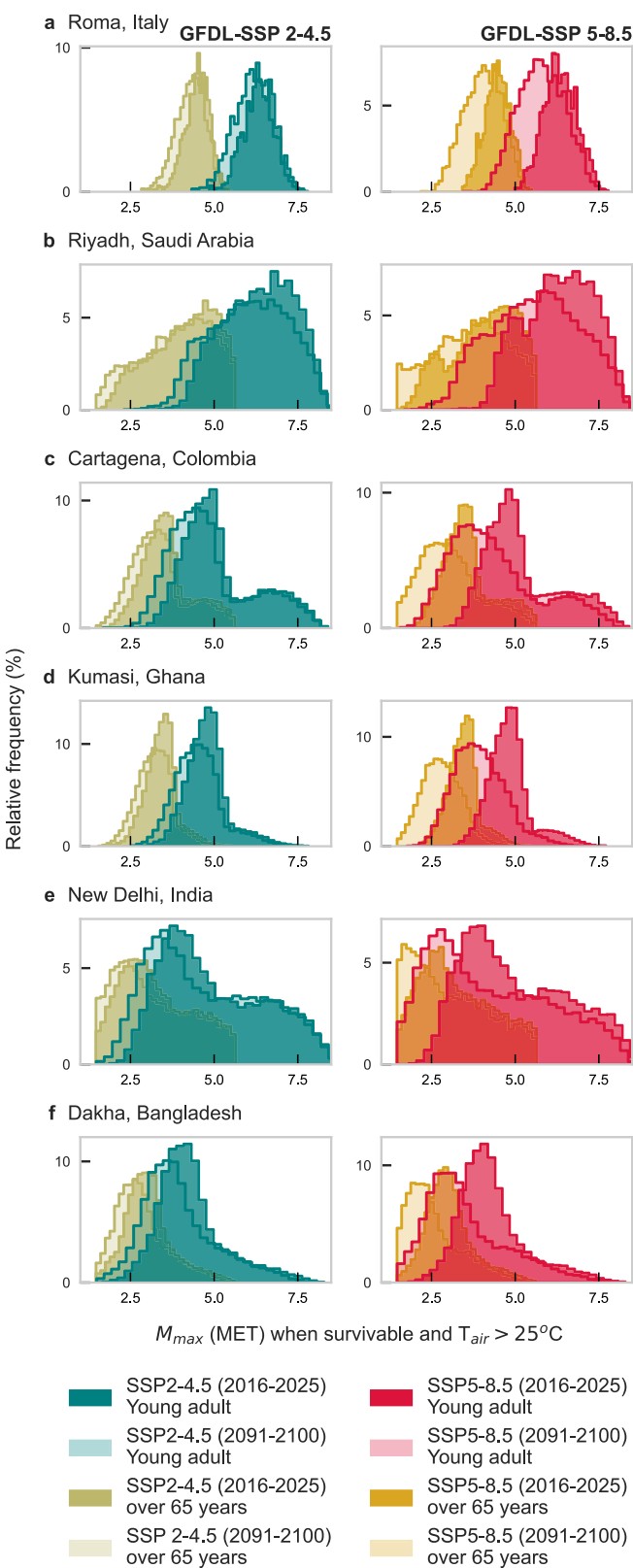

**Fig. 7 | Histograms of safe activity in six global cities comparing young and older adults as well as present and future climates.** Maximum safe metabolic rate ($M_{max}$) histograms for six selected locations (**a**) Roma, Italy; (**b**) Riyadh, Saudi Arabia; (**c**) Cartagena, Colombia; (**d**) Kumasi, Ghana; (**e**) New Delhi, India; (**f**) Dakha, Bangladesh using data from the present (2016–2026) and end-of-century (2090–2100) for young adults (teal and red) and over 65 years old (gold and yellow). 3-hourly CMIP6 data from GFDL ESM4 following SSP2-4.5 (left column) and SSP5-8.5 (right column) is used selecting the grid point closest to the city location. Note: the $M_{max}$ range truncates around a minimum value of 1.5 METs as the lower limit of activity intensity to survive but not live. Based on distinct moisture regimes, the six locations were chosen within areas that will experience the top 5% $M_{max}$ decline for young adults and overlap with the main cities in those regions. CMIP6: Coupled Model Intercomparison Project phase 6; GFDL ESM4: Geophysical Fluid Dynamics Laboratory Earth Systems Model 4. SSP: Shared socioeconomic pathway. Source data are provided as a Source Data file.

differences; if it were, the 3-hour shaded model (dotted blue line, Fig. 2a) would overlap the 35 °C $T_w$ (thick solid black) line. Instead, substantial differences persist, especially above $T_{air}$ of ~45 °C. In fact, no singular $T_w$ threshold matches the non-linearity of the physiological survival threshold because sweating restrictions cannot be captured using a fixed $T_w$ approach.

Based on the $T_w$ 35 °C exceedance[34], recent studies identify vulnerable areas around the Arabian Gulf[35], the North China Plain[55], the Ganges and Indus River basins[16], and some coastal subtropical locations. Our model indicates that heat health impacts are vastly underestimated (particularly in dry regions and for older adults) if applying the 35 °C $T_w$ threshold in current and future conditions. In arid regions projected to reach $T_{air} > 53$–55 °C, the survival of young, healthy adults in conditions otherwise conducive to minimal heat stress will be threatened. The prospect of survival for the elderly following 6 hours of exposure to $T_{air} > 46.4$ °C, irrespective of humidity, is bleak. While reducing the exposure duration from 6 to 3 hours pushes these maximum temperatures to ~54.7 °C and 48.6 °C for younger and older adults, respectively, survival limits will still be reached at much lower $T_w$ values than 35 °C. As a result, it is likely that regions characterized by low future heat stress risk in survivability/adaptability studies[12,15] (e.g., dry regions, areas with high elderly populations) will experience unsurvivable heat extremes without appropriate heat adaptation. Future work can leverage the advanced modeling capabilities and projected global demographics changes to illustrate likely future impacts using this model (see Table 3, Python Module and SM).

Assuming that people are inactive (minimal basal heat generation) is a common characteristic of any model assessing human survival in extreme heat. However, for a region to be truly liveable (or habitable), people must carry out essential activities, sometimes outdoors, even during the hottest times of the day. Hence, our liveability approach estimates human impacts beyond life or death, answering how people can live and be active in extreme heat environments without increases in $T_{core}$. Daily tasks such as writing, desk work, and typing (1.8 METs), general housework (3.3 METs), and gardening (4.4 METs) generate more metabolic heat[56]. The endogenous heat loads of occupational tasks such as digging (5.0 METs), building roads (6.0 METs), and bailing hay (7.8 METs) are even higher[56], however, people can still do these strenuous activities if they self-pace to avoid heat storage. Based on our model, a sustained activity of ~4.5–5.0 METs (e.g., dancing) is safe for temperatures above 25 °C, yet more strenuous activity is possible by lowering heat exposure, taking breaks, and/or self-pacing[57,58].

The application of our liveability model using climate projections demonstrates that under a low-to-moderate emissions scenario (SSP2-4.5), median reductions in $M_{max}$ will be modest (-0.25 METs), which more than doubles to -0.64 METs under the higher emissions scenario SSP5-8.5. These changes could represent a slightly lowered

exposure to a $T_w$ of 35 °C, $T_{core}$ would reach ~48–50 °C[54]. Yet, heat stroke death is almost guaranteed after reaching a core temperature of 43 °C[29], which aligns more closely with the approximate $T_{core}$ that would be reached after a 3-hour exposure if all remaining assumptions of the 35 °C $T_w$ model are maintained. Therefore, the translation from heat storage to $T_{core}$ between our physiology-based approach and the 35 °C $T_w$ limit is not the primary driver of their

**Table 3 | Future collaborative work among various sub-topics and disciplines**

| Sub-topic or discipline | Future work/next steps based on model capabilities |
|---|---|
| Thermal physiology | Create a catalog of specific estimates of skin wettedness, skin temperature, and maximum sweat rates across population types, ages, medications intake, and health disorders/chronic diseases. |
| | Model the impact of personal cooling strategies, such as dousing or misting skin, electric fans, and foot or hand immersion in water[78,79]. |
| | Conduct a sensitivity analysis of wind flow impacts within the liveability and survivability models, and activity velocity impacts for liveability. |
| | Model changes to behavior based on thermal exposure (e.g., Vargas et al.[80]) and other physiological attributes that affect adaptive behavior. |
| | Characterize other measures of physiological heat strain, including cardiovascular and renal strain (not only heat stroke, as modeled here). |
| | Test and model different clothing factors within the liveability and survivability models. |
| | Empirically test maximum duration of intensity safe activities ($M_{max}$) in different extreme environmental conditions and describe the changes in skin temperature and sweat rate over time and across the lifespan (e.g., refs. 39,62,81) to allow for appropriate ranges and scaling factors applied within both liveability and survivability models. |
| | Improve the steady-state version of the model, as applied here, to account for changes within the 3–6 hour exposure in the variables assumed as constant. |
| | Conduct experiments across a range of plausible biophysical parameters for uncertainty estimation and subsequent model application across different temperature/humidity combinations along the survivability and liveability threshold curves. |
| Climate sciences | Run the present model using multiple GCMs and emissions scenarios to provide an ensemble of futures, addressing various sources of uncertainty, including that from both the biophysical and climate models (e.g., Petkova et al.[82]). |
| | Obtain regional or city-specific analysis of future heat stress with downscaled, bias-corrected climate projections. |
| | Evaluate time-of-emergence for different survivability and livability thresholds using single (global climate) model initial-condition large ensembles (SMILEs). |
| | Assess the relative importance of human population trends (increase in the number of people, aging) versus climate change in future health burdens from extreme heat. |
| | Quantify the probability of exceptional heat wave and/or mass heat fatality events over different regions with different levels of climate change. |
| Public health | Determine the sources of differences between physiological models and epidemiological models to inform future model improvements. |
| | Quantify compounding and cascading risks connected to heat and health, including heatwaves compounded by wildfires. |
| | Evaluate the extent to which different urban designs would alter liveability or survivability. |
| | Leverage the new modeling capabilities along with projected global demographics to better determine impact across a diverse population for public health preparedness. |

Future work to be accomplished and tests to run connected to our new physiological modeling approach to allow for numerous analysis types and answers to larger questions.

productivity (e.g., fewer crops harvested, the need for extra workers) or decreased activity performed, with economic or health consequences[57–59]. While quantifying these consequences is beyond the scope of the present paper, this should be the focus of future research. Further, under SSP5-8.5, some places will transition from liveable to only survivable, increasing from a frequency of <0.5% of the time today to up to 7.8% by end-of-century (i.e., up to 6 months/decade, Supplementary Fig. S13) (these increases do not occur under SSP2-4.5). The sensitivity of this result to emissions scenarios and other aspects of climate change uncertainty should be explored in future work. Given the most populated global regions[60] (apart from Northern Australia) are expected to have the greatest $M_{max}$ declines, population growth will continue to increase the number of people impacted globally unless adaptive capacity increases[61].

Most striking, however, is the clear and prominent (-1.3–2.9 METs) reduction in $M_{max}$ with aging (Fig. 7). Older people inevitably become less active as they age; however, $M_{max}$ values during warm conditions in some cities, even during this decade, could constrain them to merely essential low-intensity indoor tasks such as light housework, cleaning, and washing dishes requiring only 2.0–2.5 METs.

Emerging empirical data from laboratory studies provide physiological support of our model results for young, healthy adults in shaded conditions. Wolf et al.[62] define critical environmental limits as the point at which participants enter uncompensability (which is how we define liveability) see also Table 1. The data reported by these studies[39,62] are given for $T_{air}$ from 36 °C to 50.5 °C for resting subjects (1.8 METs–filled circles Fig. 4a) and are ~0.0–2.0 °C $T_w$ higher than the liveability limits predicted by our model, likely due to lower METs here (1.5 METs). Relative to the critical $T_w$ limits for light physical activity (3.2 METs–open circles Fig. 4a) reported by Wolf et al.[62], the values

from our model underestimate $M_{max}$ by ~0.04 METs (for resting, our model overestimates $M_{max}$ by merely 0.2 METs).

Our model results illustrate the importance of accounting for specific subpopulation groups' attributes when projecting impacts from extreme heat stress. The vulnerability of older adults to heat-related illnesses is well-known[63,64]. Our survivability model identifies conditions beyond which even the fittest young adults, at best, will perish over 3 or 6 hours without access to cooling resources. This work further describes conditions that will inevitably have deadly heat stroke consequences for the elderly—a critical part of every community—while acknowledging that higher health impacts could result when considering cardiovascular heat-related effects[1,65]. Moreover, reductions in safe activity levels for young and older adults between the present and future indicate a stronger effect from aging than from warming over this century (Fig. 7).

Many of the most at-risk regions are also the most populated. While infrastructure providing artificially cooled environments may serve as a solution in some settings[66], required resources preclude this option for many, especially in lower- and middle-income countries[67]. Extreme heat may also increase the likelihood of displacement or migration of people from non-survivable and non-liveable areas[33], increasing risk for resource competition and human conflict[68]. Future research is needed to identify regions where effective heat stress adaptation measures should be prioritized and as warmer conditions arise over the coming decades[69]. Table 3 indicates future work and methodological possibilities to address further research gaps and shortcomings discussed in this study but not examined quantitively, requiring significant collaboration among disciplines[70].

In summary, this paper establishes an advanced modeling approach based on human physiology for assessing survivability and

liveability across subpopulations (younger and older female adults) and diverse climates, suitable for application with global climate model output. Among survivability assessments, we fill specific gaps around these factors across the full spectrum of temperature and humidity combinations, as compared to the common $T_w$ of 35 °C threshold, and introduce a new method to determine liveability.

Results show a vast overestimation of human limits to survival when using the 35 °C $T_w$ survivability assumption, especially for older adults and hot-dry regions. Compared to the 35 °C $T_w$, differences in physiological survival limits range from 0.9 °C $T_w$ lower (young adults, humid conditions) to 13.1 °C $T_w$ lower (older adults, dry conditions). By end-of-century, liveability declines are expected, mainly in already-populated and heat-vulnerable regions. Reductions in safe, sustained activity levels between present time and end-of-century in young and old adults indicate a stronger impact from aging on heat-health risk than from warming, thus the spatial extent and intensity of intolerable heat stress in an aging population cannot be understated. This work addresses fundamental shortcomings of common models estimating future human habitability or survivability by taking a physiological approach while opening avenues for more robust analyses (Table 3). Results and the flexible approach will advance methods in global survivability and liveability analyses under increasing heat stress. Findings underline the need for continued research efforts and investments in heat risk management, adaptive capacity, and technological innovation for personal heat protection in vulnerable global regions.

## Methods
### Overview
Our approach leverages methods of partitional calorimetry to model human heat balance[71] describing heat transfer between the human body and surrounding environment for warm conditions. Models are run across all plausible combinations of air temperature and moisture levels (e.g., RH) for warm conditions. The baseline approach builds on the general core principles of assumptions within Sherwood & Huber[12], with additional complex rational equations and population-specific inputs. The complete model, including all equations and the fundamental physiology and biophysical details, are outlined in the SM and available code. Below, we briefly explain how the full model is applied to comprehensively predict heat stroke death (survivability) and maximum safe sustained levels of physical activity (liveability) (Fig. 1).

We evaluate environments across all plausible moisture levels for warm $T_{air}$ between 25–60 °C, holding windspeed constant at 1 m.s⁻¹. Assessments are completed for indoors (or shade), where mean radiant temperature ($T_r$) = $T_{air}$, and outdoors (sun-exposed), where $T_r = T_a + 15$ °C, assuming partly sunny conditions averaged over eight midday hours[19]. Finally, we assess healthy young (18–40 years) and older female adults (>65 years). Specifics and assumptions are listed in Fig. 1 and the SM.

### Survivability estimates
Survivability estimates are provided for 3 and 6-hour of constant thermal exposure to align with current climate model outputs and past studies. The model detects increases in $T_{core}$ from an initial value of 37.0 °C to ≥43.0 °C. The $\Delta T_{core} = 6.0$ °C represents the upper limits of heat stroke[29] (however, we acknowledge that the severity of hyperthermia is variable, ranging from 41 to 47 °C[72]). Thus, this model assumes that a person cannot store more than 17.88 kJkg⁻¹, taking a typical human's heat capacity ($C_p$) as 2.98 kJKg⁻¹°C⁻¹ [73].

Therefore, the critical rate of heat storage (S) before inevitable heat stroke death during rest is 0.82 WKg⁻¹ for a 6-hour constant exposure ($S_{surv_6}$), and 1.65 WKg⁻¹ for a 3-hour constant exposure ($S_{surv_3}$). Assuming a resting metabolic rate of 1.5 METs (1.8 Wkg⁻¹), the maximum permissible net $H_{loss}$ is 0.98 Wkg⁻¹ for 6-hour and 0.15 Wkg⁻¹

for 3-hour exposures:

$$H_{loss6} = (H_{prod} - S_{surv_6}) = (1.8 - 0.82) = 0.98 \, (\text{Wkg}^{-1}) \tag{1}$$

$$H_{loss3} = \left(H_{prod} - S_{surv_3}\right) = (1.8 - 1.65) = 0.15 \, (\text{Wkg}^{-1}) \tag{2}$$

where $H_{loss}$ is the net heat loss from the skin surface to the surrounding environment and $H_{prod}$ is internal metabolic heat production. Note that $H_{loss}$ is a function of weight and thus would vary across populations. $H_{prod}$ is equivalent to metabolic rate (M) because $W_k = 0$. A person is also assumed nude for the survivability assessment (thus, dry and evaporative heat transfer clothing resistances are 0 m²°CW⁻¹), in line with assumptions made by other survivability models[12].

Our approach models evaporative restrictions to heat loss, which can exist due to three factors (see Supplementary Fig. S1): (1) high environmental humidity (or biophysical evaporative heat loss limit ($E_{max_{env}}$); (2) the physiological capacity to saturate the skin surface in high humidity environments due to a limited maximum skin wettedness ($\omega_{max}$); and (3) by the maximum rate at which sweat can be produced ($S_{max}$). These are factored into both survivability and liveability analyses.

For survivability, we follow the framework in Supplementary Fig. S1 to determine whether a person can survive heat stroke based on three questions, with a potential for four outcomes given individual characteristics (e.g., differing age and size). Thus, the algorithm determines survivability (as a dichotomous variable: yes/no) and assigns outcomes based on combined environmental and physiological restrictions. Based on this framework, a person will (see zones in Fig. 2):

1. Survive while remaining within sweating limits.
2. Survive despite exceeding sweating limits.
3. Not survive because the environment restricts heat loss too much (in high humidity).
4. Not survive because the required sweat rate is not possible (in low humidity).
5. Not survive due to both critical environmental heat loss restrictions (3rd argument) and a required sweat rate that is not possible (4th argument).

### Liveability estimates
Liveability is the maximum metabolic rate ($M_{max}$) that can be generated before S ≥ 0, or sustained compensable heat stress, where M = $H_{prod}$. The $M_{max}$ value indicates the sustained activity levels (intensity but not duration) possible without unchecked rises in $T_{core}$ (i.e., uncompensable heat stress[74]) within a given steady-state thermal environment. This definition assumes that people will self-pace their maximum level of physical activity over a prolonged period to attain S = 0, as well as constant sweating and skin temperature. $M_{max}$ is a continuous variable (in Watts), which we convert to energy expenditure in METs for a simpler interpretation. A clothing insulation value of 0.36 clo is used (light shorts and cotton T-shirt) for liveability. Full details are in the SM.

### Application of liveability model using climate models
To illustrate advanced model capabilities, we explore the distribution of liveability in present and projected future conditions in regions globally (as outlined above and in SM). The Coupled Model Intercomparison Project phase 6 (CMIP6) provides state-of-the-art GCMs projections under various warming scenarios[75]. We utilize CMIP6 data from GFDL ESM4 and MPI ESM1.2 (-1° × 1.25° atmosphere/land grid) following SSP2-4.5 and SSP5-8.5 (Shared Socioeconomic Pathway). SSP2-4.5 represents a middle-of-the-road emissions scenario, which may be likely considering existing net-zero commitments[76], and SSP5-8.5 represents a high emissions scenario. As inputs, we use 3-hourly

near-surface air temperature, specific humidity, and surface atmospheric pressure for two 10-year global snapshots—present (2016–2025) and future (2091–2100)—to construct global maps of projected changes over land and detail $M_{max}$ in select cities. A grid cell was considered land if it contained >45% land cover; otherwise, analyses were not performed.

We used two-sided Mann-Whitney U tests to detect significant projected changes in $M_{max}$ between present (2016–2025) and future (2091–2100) decades. Here, we test differences in median $M_{max}$ (50th percentile of distribution) using a significance level of $p < 0.05$ (Fig. 6). Six cities within the global top 5% of expected declines of $M_{max}$ were selected to display deceased $M_{max}$ in statistical distributions (Fig. 7): Roma, Italy (a); Riyadh, Saudi Arabia (b); Cartagena, Colombia (c); Kumasi, Ghana (d); New Delhi, India (e); and Dakha, Bangladesh (f). The Mann-Whitney U tests were used because $M_{max}$ is a continuous variable, not always normally distributed. We evaluated liveability changes solely for shaded warm/hot weather conditions ($T_{air}$ >25 °C).

Our analyses focus on exploring uncertainties associated with the physiological model; future work will complete a more comprehensive climate change impact analysis using many GCM simulations to consider uncertainties due to model structure, internal variability of the climate system, and emissions scenario. GFDL ESM4 and MPI ESM1.2, like any climate models, possess biases in their simulation of mean climate and the diurnal cycle, and do not capture details of urban landscapes[77]. Future impact assessments might consider bias correcting and downscaling the GCM output prior to inputting in the physiological model.

### Reporting summary
Further information on research design is available in the Nature Portfolio Reporting Summary linked to this article.

## Data availability
The model output data generated in this study are provided as source data files. Input data included all plausible combinations of air temperature and moisture levels for warm conditions, which we used for generating survivability wet-bulb curves and liveability analyses; therefore, the only external data comes from CMIP6 data. The input data for the model for the personal profiles custom-built for this application are provided in the methods, supplementary information, and Zenodo repository. The CMIP6 data were downloaded from https://esgf-node.llnl.gov/search/cmip6/. All the source data for figures are provided in the Zenodo repository as well and with this paper as a Source Data file, including the wet bulb temperature matrices from the combinations of air temperature and humidity tested. Source data are provided with this paper.

## Code availability
The model code was developed using Python 3.10.9 and authors thank the teams behind this open-source project, as well as NumPy (v1.23.5), Matplotlib (v3.7), Xarray (v2022.11.0), Pandas (V 1.5.3), Cartopy (V 0.21.1), and MetPy (1.4.1) developers. Custom codes with the model developed for this study and tutorials to reproduce survivability and liveability temperature-humidity matrices are available via Zenodo data repository (https://doi.org/10.5281/zenodo.10020136).

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

## Acknowledgements

GEG and JKV acknowledge funding from the National Science Foundation (NSF) Award #CMMI-2045663. O.J. acknowledges funding from the National Health and Medical Research Council (NHMRC) Investigator Grant (2021/GNT2009507). We acknowledge the World Climate Research Programme, which, through its Working Group on Coupled Modeling, coordinated and promoted CMIP6. We thank the climate modeling groups for producing and making available their model output, the Earth System Grid Federation (ESGF) for archiving the data and providing access, and the multiple funding agencies who support CMIP6 and ESGF. We are grateful to Jared Sexton for his help with obtaining the CMIP6 data and Haley Staudmyer for testing the code and $T_w$ calculation troubleshooting. We thank Colin Raymond and Alex Goodman for sharing improved Python code to calculate $T_w$. We would also like the Noll Lab at Pennsylvania State University, particularly Larry Kenney and Daniel Vecellio, for helpful discussions around heat stress and evaporative heat loss. Finally, we are grateful to Ariane Middel at ASU for providing GIS mapping expertise and direction for the MRT modeling methods used in the current paper and Stevan Earl for helping with data management good practices to publish the code.

## Author contributions

All authors conceptualized the study. J.V. acquired funding and supervised the project. O.J., C.B., G.G.E., and J.V. performed model development and advancements. Climate data were obtained by J.B. G.G.E. developed the model code and performed data analyses with guidance from J.V. and O.J. J.B. and K.L.E. led interpretations of climate projections. J.V. led manuscript development and writing; all authors helped with interpretation, writing, and feedback. All authors approved the final manuscript.

## Competing interests

The authors declare no competing interests.
