## [Peer Review File · Nature Communications]

A physiological approach for assessing human survivability and liveability to heat in a changing climateREVIEWER COMMENTS

Reviewer #1 (Remarks to the Author):

Review of “ A physiological approach for assessing human survivability and liability to heat in a 2 changing climate” by Vanos, et al.

The authors integrate a physiological model of heat stress with climate projections, to enumerate how, survivability and livability at various activity levels relate to temperature humidity contributions. They also consider people over 65, and shaded vs. sunlight conditions. The paper is generally very well-written, and with minor-to-major revisions I believe it will ultimately be an important contribution to the literature and worthy of Nature Communications. There are however, some general and specific (line by line) concerns/comments, I have, which I outline below respectively.

General/Major Comments

1. The single biggest question, which I don't feel particularly well-suited as a climate scientist to answer, is how much trust to put in their this study, based on a single physiological model and the Wolf reference. Everyone at this point knows at this point that a lot of people will be dying long before 35C wet bulb (see Horton et al. 2021 for example on humid heat and habitability) but the extent of your downward revisions under high dry bulb low RH, and the absence of downward revision for high RH and 'low' dry bulb T are...striking, and major departures from most prior work in a field that has built up a fair number of papers in a hurry. Here I should note that I myself have been burned by reviewers with blinders on, who told my co-authors our results, ~ 'weren't plausible' without pointing to anything more than their intuition. But I find myself feeling similarly about some aspects of these findings. I would feel better if these results were compared to other physiological models, results from (even critiqued) epidemiological studies, etc. Most fundamentally, the authors don't really 1) discuss that their findings at least to my mind, don't align with experience and 2) offer explanations, based on what their model may be missing.

As examples of the former, the model seems insufficiently sensitive at low RH and high dry bulb temperature extremes. What for example should we make of all the outdoor, sun exposed elderly people around the world who are surviving and performing basic activities at dry bulb T and RH combinations your findings suggest should not be possible? Or in fig 3, all the 18-40 year olds who are able to do strenuous (running) activities outdoors at 86F and near-zero humidity in the absence of sun? Doesn't one find 18-40 year olds able to actively play/dance in the sun at 86 and zero humidity?

These points seem even more problematic when one notes that the model used by the authors neglects other health issues that can limit sweating, and assumes nudity. How can this be explained?

The models also seem too sanguine about high RH and lower dry bulb temperatures. Does evidence really suggest that 93F and 100 percent RH would be survivable for a 65+ in the sun?? Or that they could do light walking in the sun at 79F and 100 percent humidity? These findings seem questionable too, although here it is easier to invoke things missing from the model perhaps, like pre-existing health conditions.

2. (And related to '1' above): Word count restrictions notwithstanding, in general more caveating and discussion are needed. For example, mention your assumption of very weak wind. While I am not asking you to model other wind speeds, it seems insufficient to not mention your assumption in the main, and how it could impact your assumptions about max cooling through sweating.

In general, the paper needs to talk more about smooth gradients, for example the idea that the 65 age cutoff you use is not a perfect binary. What about people with health conditions? What about people with different constraints/goals in their activities (Horton et al. 2021)? Adaptation options like putting cold water on yourself? Note that I am not suggesting new analyses here. Rather that there is a chasm between the needs you articulate in the introduction, and the nice additions you address through this nice study. In the discussion, circle back to remind us how your study helps fill in some of the missing pieces, but a chasm remains before anyone is able to fully model/understand all the factors that influence the relationship between humid heat and mortality/liveability.

Most fundamentally, you need to be much clearer about which aspects of the gaps in our understanding/prior work your analysis addresses, and which ones it doesn't. For example, lines 49-52 read almost as legerdemain, that needs to be cleaned up. I think you can still make your overall point, even if it means adding a sentence or two, in a way that acknowledges that some epidemiological/econometric approaches (even with projections) have included humidity. See my comments above about how you need to engage epidemiology/econometrics/empirical approaches a bit more. This is one of the major branches of analysis in assessing heat impacts on health, so it needs significantly more attention here. This can help strengthen the case that a reader should instead trust your single physiological model result!

I think the Introduction is generally strong—well written, especially when describing gaps and limitations in prior work. That said, I needed a little more info before the paragraph on lines 88-92. Despite excellent text before this paragraph, the 43C threshold still comes seemingly out of nowhere in the first sentence. And then in the second sentence we hear about maintaining core temperature, which leads one to wonder about the gray area between these two extremes. I don't question the use of these two measures—we just need some added context before you introduce these two specific 'lines in the

sand' after talking too us earlier about all the nuances that can render lines in the sand blurry (pre-existing health conditions, adaptations, etc.).

Again, your summary of what is needed, and the importance of more nuance in these studies is strong. It is just odd that you aren't more precise about which of these many pieces you all chew off in your analysis. Without a close read, one could easily assume your study addresses ALL the shortcomings of prior studies you have just (nicely) elucidated, rather than just a couple (important ones).

Along similar lines, the critiques of prior work on lines 66-67 seem fair, but seem to promise more than you deliver. For example, in contrast to prior authors, do you consider body size, or more than a single (age 65+) 'restriction to thermoregulation'? Why not say that you don't?

3. I am trying to figure out how in the extremes of high T and low RH the downward revisions of the danger thresholds relate/don't relate to actual weather conditions (as opposed to theoretical or lab experiment-driven conditions). Once temps get really high (e.g. 115+ F) on your figures, there currently aren't any RHs of 40 percent, for example. Along similar lines, does any literature suggest that anything like 35C wet bulbs occur or will occur at low RH? In a sense my point is that not only does this paper focus on a single physiological/human model, not only does it largely punt on describing humid heat epidemiology, but it also largely punts on describing the climatology of observed heat and humidity. It is critical to know which combinations of T and RH are experienced and likely to be experienced. As a bare minimum revision, can we see reported combinations of T and RH somewhere, perhaps added to some of the existing figures 1-4?

4. One GCM, one RCP is a problematic. For Nature Communications, I think you need to include more models in the main, and probably another RCP in the Supplement. Note that these additions would not starve the follow up study you refer to about sources of uncertainty, in a field with so many possible permutations (Petkova et al. 2014, for example.)

Line by line comments

Liability' in title to 'Liveability'

Abstract, second sentence: move the last 9 words (perhaps further shortened) up to earlier in the sentence. As written, it seems like they only apply to livability not survivability.

Introduction

Is the term 'heatstroke death' sufficient here? (I am wondering if there are other kinds of heat related deaths you mean to include that are not captured by the term 'heatstroke death')?

Penultimate paragraph—clarify that you mean an individuals aging, rather than the average age of humans (or more to the point the number and age of older people) going up in the future

'3.2' at first use in heading to 2.2'

'3.3 at first use in a heading to 2.3'

I feel Figures 1 through 4, while all excellent, are so similar stylistically and content-wise similar as to argue for making a hard decision to move at least one of them (possibly two of them) to the supplement.

Line 273: 'Plan' to 'Plain'

Figure 3: Give us a better sense, through examples in the text of what levels of activity different values of 'Met' correspond to. Figure 3 is nice in this regard, but say more in the main text.

Also do the icons, like running, assume no rest during 3 hours?

Figure 5a: why are some areas showing reduced MET?

Figure 5b: worth exploring why Bangladesh has the lowest values. I think of this area as closer to high RH low dry bulb than many of the other humid heat hotspots around the world like the Indus Valley for example. Given your findings (high dry bulb at low RH is more dangerous than most researchers have realized) this is somewhat counterintuitive. Can you look into some explanations? (Maybe GCM resolution, maybe the sheer number of days per year is a more dominant term in your model [maybe Bangladesh has a lot of pretty oppressive days, rather than having the individual days that are the most extreme], etc.).

Figure 6 caption: 'Dix' to 'Six'

Figure 7: I think this should be much earlier in the paper.

Figure 8 caption: Is 'wittedness' intended? Especially if you are constrained in your number of Figures, this is a very logical figure to move to the Supplement.

Methods 4.4. did you bias correct for cities? If not, it is OK, but you should say so.

References

Horton, R. M., de Sherbinin, A., Wrathall, D., & Oppenheimer, M. (2021). Assessing human habitability and migration. *Science*, 372(6548), 1279-1283.

Petkova, E. P., Vink, J. K., Horton, R. M., Gasparrini, A., Bader, D. A., Francis, J. D., & Kinney, P. L. (2016). Towards more comprehensive projections of urban heat-related mortality: estimates for New York City under multiple population, adaptation, and climate scenarios. *Environmental health perspectives*.

Reviewer #2 (Remarks to the Author):

The manuscript focuses on assessing survivability and livability by applying physiological and biophysical principles. The authors suggest that the 35°C T_w threshold underestimates the risks for older adults. They also suggest that the risk continues to increase in a changing climate.

I have two major concerns with the manuscript. First, the authors use only a single GCM and a single scenario that fails to capture the range of future climate uncertainties. Secondly, the authors extract city-level data from a pretty coarse resolution GCM. The majority of GCMs in CMIP6 are still pretty coarse resolution including the GFDL-ESM4 model that the authors have used. Using GCM data to draw such conclusions can be misleading given that it fails to capture various city-specific features and will have substantial biases. From the title of the manuscript, it looks like the authors are trying to evaluate the survivability and livability in a changing climate, and therefore, using single coarse resolution GCM is not justified. A minor comment- it should be "livability" and not "liability" in the title.

Reviewer #3 (Remarks to the Author):

OVERALL:

The goal of the paper by Vanos et al. was to improve upon the modeling of survivability and livability limits of humans exposed to increasingly hot environments due to climate change. In many ways the authors succeed in their goal of advancing upon earlier modeling efforts. The authors, in particular, extend survivability and livability limits to very hot and very dry environments and attempt to show differences between young and old. Other advantages are mentioned (contingencies for disease states, acclimatization status, body size, fitness, etc.) but not described. The work submitted is in many ways masterful, but there are numerous potential problems with the many assumptions made and as a result the interpretations also appear problematic – even incorrect. A model such as the UTCI, validated against empirical data (such as Brode and Kampmann, 2023), may be a better approach? Questions and comments follow for consideration.

MINOR:

Line 1: Title - liability should read livability.

Lines 183, 191: A decrease in M_{max} of 0.2-0.26 METS does not seem like an appreciable difference (10-20% can be the measurement noise for METS). Can the authors better contextualize how this matters to human health, economic burden, etc.?

Line 372: Did you mean reference #30 for this statement? Reference #67 I understand.

Lines 389-403: It would be helpful for those interested to introduce the average height and weight used to arrive at the 1.60 m² (young) and 1.78 m² (old) (supplementary information) values that appear to have been used for the calculations represented in all the tables and graphs. Were these population median values for height and weight in young and old populations?

Line 406: It does not seem reasonable to use a nude body for survivability (how many people would be nude?) but minimal clothing for livability. I understand wanting to compare your results to previous work, but the buck should stop somewhere? Or consider using nude estimates for livability as well?

References: There are a number of references with errors in them. Please review and edit.

Table 2: The lower T_w limit at lower humidity seems strange without the context of the very high accompanying air temperatures.

Figures 1-4: Where are environmental conditions projected to rise to 50 degrees C or more to make these projections less theoretical and more meaningful?

Figure 6: Can the authors explain the y-axis probability?

Supplemental Material and Model Assumptions:

Equation 20: How is S_{max} determined? Is it simply E_{req} ?

MAJOR:

Line 389: The premise for using 43 degrees C for survivability is justified by citing the range (41-47), but 40.8 degrees C was the average exertional heat stroke temperature observed in more than 100 military cases (see Figure 5-3 in:

(https://armypubs.army.mil/ProductMaps/PubForm/Details.aspx?PUB_ID=1024722). Is 43 degrees C more applicable for passive heat stroke? Should a distinction be made between passive and exertional heat stroke? Which are you modeling, specifically? Both? A more justified value ~41 degrees C could drastically change your results and interpretations.

Lines 432-436: I am not sure I understand $S = 0$ in this explanation. Heat storage could and would take place to elevate body temperature (from 37 degrees C) to a new but stable baseline (e.g., 37.5, 38.0, 38.5,...). Why must no heat storage be assumed (not realistic)?

Supplemental Material and Model Assumptions:

1. Is airflow 1 m/s used in both survivability and livability scenarios? In Figure 3, a MET value of 2.5 (walking) arguably creates airflow around a person equal to ~1 m/s, but jogging at 7.5 METS would be more like ~2.25 m/s. Riding a bicycle at 5 METS would produce an airflow of ~5 m/s, completely ablating the insulative air boundary layer. The under-estimation of airflow will have important effects on your interpretations that rely so heavily on evaporative cooling.
2. Mean T_{sk} is set to 35 degrees C. Though it may align with other publications, mean T_{sk} can be easily 36 or 37 degrees C in very hot air (40 to 50 degrees C), especially when globe temperature is higher still. Some military models use 36 degrees C or establish a scaling factor for the best gradient. This too will potentially impact your interpretations and calculus as dry heat gain is somewhat exaggerated.
3. The skin wettedness values employed of 0.50 to 0.65 seem too low, especially 0.50 (e.g., Candas et al., 1979; Ravanelli et al., 2018). This makes a huge potential difference to E_{maxlim} . Can the authors provide justification for such small values? If 0.65 to 0.85 were used, how would this impact your results? I suspect greatly.
4. There is a major emphasis on how age impacts the results of the model, but little explanation is given for why. Reference #15 is cited for age negatively impacting E_{maxwet} . However, while reductions in vasomotor and sudomotor function have been convincingly demonstrated to occur with aging, translation to heat balance on the whole body level is less clear. For example, men 45 years older than younger subjects matched for body mass, height, and body surface area had 50% lower local sweating rates, even though E_{req} in W/m^2 was similar (Schmidt et al., 2022). Others have quantified an age-related reduction in thermoeffector heat loss potential of ~4% per decade between ages 18 and 70 (D'Souza et al., 2020). How does this compare with your model? On the whole body level, small or non-significant differences in heat balance between young and old are reported (Stapleton et al., 2014). Importantly, the Stapleton study was conducted at 40 degrees C and 15%rh at metabolic rates ranging from 300 to 500W. Your results (lines 166-169) seem to suggest that this should be impossible for older adults (65 yoa in Stapleton et al., 2014).

5. Would use of the UTCI model be a better approach? It has recently been validated against empirical data. Please see: Bröde P, Kampmann B. Temperature-Humidity-Dependent Wind Effects on Physiological Heat Strain of Moderately Exercising Individuals Reproduced by the Universal Thermal Climate Index (UTCI). *Biology (Basel)*. 2023 May 31;12(6):802.

Overall, it seems that a small change in any one of many model assumptions could significantly alter your results, interpretations, and conclusions.

Point-by-point Response to Reviewers Comments:

REVIEWER COMMENTS

Reviewer #1 (Remarks to the Author):

Review of “ A physiological approach for assessing human survivability and liability to heat in a 2 changing climate” by Vanos, et al.

The authors integrate a physiological model of heat stress with climate projections, to enumerate how, survivability and livability at various activity levels relate to temperature humidity contributions. They also consider people over 65, and shaded vs. sunlight conditions. The paper is generally very well-written, and with minor-to-major revisions I believe it will ultimately be an important contribution to the literature and worthy of Nature Communications. There are however, some general and specific (line by line) concerns/comments, I have, which I outline below respectively.

Response: Thank you for these comments and helpful overview.

General/Major Comments

1. The single biggest question, which I don't feel particularly well-suited as a climate scientist to answer, is how much trust to put in their this study, based on a single physiological model and the Wolf reference. Everyone at this point knows at this point that a lot of people will be dying long before 35C wet bulb (see Horton et al. 2021 for example on humid heat and habitability) but the extent of your downward revisions under high dry bulb low RH, and the absence of downward revision for high RH and 'low' dry bulb T are...striking, and major departures from most prior work in a field that has built up a fair number of papers in a hurry. Here I should note that I myself have been burned by reviewers with blinders on, who told my co-authors our results, ~ 'weren't plausible' without pointing to anything more than their intuition. But I find myself feeling similarly about some aspects of these findings. I would feel better if these results were compared to other physiological models, results from (even critiqued) epidemiological studies, etc. Most fundamentally, the authors don't really 1) discuss that their findings at least to my mind, don't align with experience and 2) offer explanations, based on what their model may be missing.

Response: Thank you for this comment. The striking downward revision under the dry conditions is due to our model's ability to account for the limited capacity of humans to secrete sweat and, thus, support evaporative cooling, wherein people can only sweat so much per hour. Notice that in the T_w 35°C model, the sweat loss is unlimited. In very hot and dry conditions, the prospect of survival is overestimated because the 35°C T_w model operates solely on the assumption that the climate (humidity) limits evaporative heat loss potential rather than the physiological capacity to secrete sweat, which is well established in the physiological literature.

Many climate researchers have used the 35°C wet bulb approach. Still, it has a minimal physiological basis, and this concern is one of the reasons we set out on this study—to demonstrate that the assumption is very narrow and misses important aspects about people, behavior, and exposures, overly simplifying the complexities of the human body (see Vanos et al., 2020). Our approach here uses a more complex and biophysical approach that accounts for physiological principles (not a pure physiological

model) to look at all the avenues of heat exchange as it relates to the person (including thermoregulatory impairments), the specific conditions, and activity, among other factors that this model would allow us to adapt in the future. The estimates are not a matter of opinion, nor are they dependent on the data reported from a single study, but are based on a rational model. Assuming a single threshold of wet-bulb temperature doesn't allow for any of these considerations, and thus the prolific use of it—solely because it is simple to apply in climate models—is worrisome.

It may be that we do not clarify the following well enough within our paper. The main aim is to demonstrate the differences in potential survival and livability outcomes if such a simplistic approach (T_w of 35) is substituted with a model based on biophysical principles of human-environment heat exchange and

Therefore, we clarify these two issues in the paper on Lines ~90–93.

“Here, we demonstrate a new approach using physiological principles that align with human thermal responses to heat (heat strain) to begin overcoming simplified approaches that miss important physiological and behavioral factors of humans in the heat.”

Second, to address #1 and #2 in the comment, in the introduction and the discussion we have clarified why the findings from the 35°C wet-bulb temperature threshold studies do not align with more realistic scenarios (e.g., people are not always in the shade and laying down; ageing affects thermoregulation). real-world experiences. We also offer explanations as to why these issues arise and what the 35°C wet-bulb temperature model might be missing to offer an improved approach.

The following text is added on line 80–82:

Within the introduction: *“The 35°C threshold approach cannot capture complexities and personal characteristics affecting human thermoregulation (e.g., body size, activity levels, or physiological restrictions (such as sweating) to thermoregulation^{6,19}), which can cause large errors in estimations.”*

Within the discussion, the first paragraph of section 3.1 emphasizes the constraints of the popular T_w of 35°C threshold model.

Also, note that we have not used results from Wolf et al to develop our model. Rather, the Wolf data are merely provided to illustrate some context using empirical data from an physiological study, which aligns quite well with our liveability model. Further, the Wolf et al. results also support the notion that the T_w value for thermal compensability (our definition of liveability) is much lower than 35°C under dry conditions. Note that the Wolf study defined the critical limit as the point at which people tip into uncompensability (which is our definition of liveability). In contrast, we permit a certain rate of heat storage for survivability, allowing a possible range of T_{core} increase until 43°C within a 3H or 6H timeframe. We have made these factors more evident in the discussion on Lines 362–364:

*“Emerging empirical data from laboratory studies provide physiological support of our liveability model results for young, healthy adults in shaded conditions. Wolf et al.⁶³ define critical environmental limits as the point at which subjects enter uncompensability (which is how we define liveability) see also **Table 1**”*

We have also included the Horton et al. (2021) paper as a citation in a couple of places to support discussion points, and additional physiological models cited within the discussion and Table 3.

As examples of the former, the model seems insufficiently sensitive at low RH and high dry bulb temperature extremes. What for example should we make of all the outdoor, sun exposed elderly people around the world who are surviving and performing basic activities at dry bulb T and RH combinations your findings suggest should not be possible? Or in fig 3, all the 18-40 year olds who are able to do strenuous (running) activities outdoors at 86F and near-zero humidity in the absence of sun? Doesn't one find 18-40 year olds able to actively play/dance in the sun at 86 and zero humidity?

Response: Thank you for this comment, as it helps us see where we may not be adequately explaining the differences between survivability and liveability concepts. For Figure 3 (now Figure 4), the results represent **liveability**, which is the maximum SAFE internal metabolic heat production/physical activity that a person can generate without a sustained positive rate of heat storage in the prevailing environment (or without exceeding the maximum rate of heat dissipation possible from the skin to the surrounding environment). As such, above this threshold, the body will continuously store heat energy inside the body even with the body attempting to maximally shed heat through evaporation, and core temperature will continually rise. While rises in core temperature may be manageable over a short duration and not pose a health threat, activities at levels above M_{max} are not sustainable for a prolonged period due to a progressively increasing heat stress risk (due to being in uncompensable heat stress). For example, while dancing at 30°C with very low humidity in direct sunlight may be safe for relatively short durations (e.g., ~30 mins), continuing to conduct this activity under these conditions will eventually pose a health threat because of the progressive core temperature increases. The more an activity exceeds M_{max} for a given combination of temperature and humidity, the faster the core temperature will rise (and, therefore, the shorter the safe duration). Indeed, an 18–40-year-old can safely dance or run at temperatures below about 35°C with low humidity for longer durations; higher humidities would result in them being able to do less at the same temperature.

We hope this helps clarify. Because of this question, we think the interpretation of the graphs can be improved. Now for each graph in Figure 3, we have described the areas that are safely sustained (or “safe sustained activity”) for the given level of activity indicated by color, areas that are survivable but not liveable (no activity possible without continuously storing heat internally), and the area of the graph that is unsurvivable. We have also updated the caption on Figure 4 (and the similar supplemental figures) to read:

“Figure 4: Liveability estimates based on maximum safe metabolic rate (M_{max}) that a person can generate without a sustained positive rate of heat storage even with a maximal thermoregulatory response. Results are presented across a range of air temperature and relative humidity for younger (a, c) and older adults (b, d) in shaded (top) or sun-exposed (bottom) steady-state environments. The 3-hour survivability line is shown in purple; constant T_w values are shown by the solid black lines until 37°C to avoid unrealistic conditions, with $T_w=35°C$ shown by the thick black line. Activities by MET level range from no activity (sitting ~1.5 METs), to housework (~3.0 METs), dancing (~5.0 METs), and heavy lifting (~7.0 METs). The hatched area indicates conditions that are survivable but not livable (i.e., people cannot increase their activity without continuously storing heat inside the body, which will lead to a continuous rise in core temperature, but heat stroke death after a 3-hour exposure would not occur). Icons indicate MET-equivalent activities according to Ainsworth et al.⁵⁶ Circles indicate critical T_w limits reported by Wolf et al.⁶⁵ for minimal (~1.8 METs—filled circles) and light physical activity (~3.2 METs—open circles). Note: 1 MET = 58.1Wm⁻², with 1.0 METs (~58 Wm⁻²) corresponding to complete rest. See **Figure S6-7 for vapor pressure and specific humidity on y-axis, respectively. Icons provided by Icons8 (<https://icons8.com>).”**

Further, in response to this comment, also we have tried creating a clearer definition of liveability in Table 1 and the introduction.

These points seem even more problematic when one notes that the model used by the authors neglects other health issues that can limit sweating, and assumes nudity. How can this be explained?

Response: We do not neglect health issues; indeed, we highlight the importance of their representation in these kinds of studies. However, we cannot address all such health issues and the immense complexity of the human body in one paper. Given that sweating impairments due to age are robust and have been well-documented in the literature, we choose to model in this current paper that constraint in thermoregulatory response. In the meantime, physiologists are still working to characterize the effects of health disorders on sweating, and thus we do not advance the model as far just yet to avoid speculation (yet the model will be able to account for this in the future as research progresses). See the new Table 3.

We only assume nudity for the **survivability** estimations, as this is where one would wear as little as possible and do nothing. Additionally, as nudity is the assumption for the $T_w=35^\circ\text{C}$ approach, we can better compare our results with their approach (this is stated already in the main methods). Conversely, for **liveability**, where we estimate maximum safe physical activity, we assume people are wearing light clothing ($\text{clo} = 0.36 \text{ clo}$, light shorts and cotton T-shirt). These details are provided in the supplemental materials, and we have also added them to the main methods:

“We also use a clothing insulation value of 0.36 clo (light shorts and cotton T-shirt) in the liveability analysis.”

Future work can evaluate the sensitivity of clothing on liveability.

Finally, to help ensure these distinctions are clear, we have added this information to Figure 1 (which uses to be Figure 7).

Radiation Conditions	Indoors/Shaded		Outdoors/Sun-exposed	
Population Type	Young Healthy Adult (18–40 years)^	Older Adult (>65 years)^	Young Healthy Adult (18–40 years)^	Older Adult (>65 years)^
Survivability Assessment	Test: Conditions that would elicit uncompensable heat stress (UHS) leading to inevitable body heat storage & rising T_{core} to result in heat stroke in 3 or 6 hours.** Conditions: All combinations of 25–60°C T_{air} , 0–100% RH, constant wind (1m/s) Assumptions: Nude, laying down, remain hydrated.			
Liveability Assessment	Test 1: Maximum metabolic heat production (M_{max}) that can be physiologically compensated without a sustained continuous rate of heat storage. Test 2: Conditions that would elicit UHS but not a deadly T_{core} increase for a 3-hr exposure (survivable but not liveable) at rest. Conditions: All combinations of 25–60°C T_{air} and 0–100% RH, constant wind (1m/s) Assumptions: Clothing resistance of 0.36clo, remain hydrated.			
Future Liveability	Test: Change in M_{max} between contemporary (2016–2025) and future (2091–2100) climates (when $T_{air} > 25^{\circ}C$, constant 1m/s wind). Assumptions: Clothing resistance of 0.36clo, remain hydrated, indoors/shaded.			

^Assumptions on population characteristics for young and old female adults, respectively: average female body sizes by age (1.60m² & 1.78m²), constant skin wettedness (0.65 & 0.85), and maximum sweat rates (0.51 & 0.75L/hr). **Uncompensable heat stress with limited heat storage.

We think it's also worth pointing out that while we make assumptions, our model represents a significant improvement over the widely used T_w of 35°C by accounting

1. Physiological constraints to sweat production in hot dry conditions
2. Sweating restrictions due to ageing, which are the most wide scale thermoregulatory impairment in society
3. Sun exposure
4. Different morphology between different ages
5. A physiologically plausible core temperature threshold for heat stroke
6. The ability to behavioral adapt by reducing metabolic rate (in our liveability analysis)

The T_w of 35°C doesn't account for any of these. We have listed these points in the section titled: **“Considerations of New Model Estimating Physiological Survivability Limits and Liveability”**

The models also seems too sanguine about high RH and lower dry bulb temperatures. Does evidence really suggest that 93F and 100 percent RH would be survivable for a 65+ in the sun?? Or that they could do light walking in the sun at 79F and 100 percent humidity? These findings seem questionable too, although here it is easier to invoke things missing from the model perhaps, like pre-existing health conditions.

Response: If we focus on survivability for >65 years of age (Figure 2d), then yes, a person >65 years wearing no clothing and laying down not moving sun-exposed (which is partial sun in this case), light air flow, having full access to water could survive at 90°F and 100RH for 6 hours. According to the T_w of 35°C approach, a similar situation shows that any person could experience 95°F and 100 RH for 6 hours and survive without reaching heat stroke conditions (although cardiovascular strain would be very high). Notably, these are the thermal environments in which the two approaches agree relatively well, although the T_w of 35°C approach doesn't account for radiative heat exchanges. We also note in the

discussion that such a person, based on our model results, would likely move to an indoor or shaded location if they had access, which would be a better situation for them (Figure 2b).

Further, it is important to emphasize that our survivability model is using the same definition as Sherwood and Huber, i.e., survival from the perspective of heat stroke (critical rise in T_{core}). We do not account for the prospects of survival due to the heat-related outcomes (e.g. CV strain)

However, the second scenario provided in this comment (which is focused on liveability) suggests light walking in the sun at 79°F and 100% humidity (Figure 4). In this case, that person who is 65+ can indeed do light walking in the sun safely (assuming full replenishment of water and constant sweating) without a sustained positive rate of heat storage, which is similar to our response to the question above about Figure 4. We think we clarified the liveability definition, improved the annotations on the liveability graphs in Figure 4, and updated the caption (see above).

We also recognize some very extreme values used to express the full range of T_{air} and RH, and we have responded about this in comment #3 from reviewer three below.

2. (And related to '1' above): Word count restrictions notwithstanding, in general more caveating and discussion are needed. For example, mention your assumption of very weak wind. While I am not asking you to model other wind speeds, it seems insufficient to not mention your assumption in the main, and how it could impact your assumptions about max cooling through sweating.

Response: Thank you for this comment. Ensuring our assumptions are clear is very important. We have worked through all comments herein to ensure that everything is clearer and specifically make better use of **Figure 1** (pasted above) to indicate the windspeed assumption (and other key assumptions) used. All of this is also explicitly outlined in the methods, stated in the discussion, and future necessary research on windspeed is now in the new Table 3 too. Indeed, more cooling could occur through sweating as long as the physiological limitation to sweating are not exceeded. However, we feel that while of course the present model is subject to several assumptions, it is a major improvement on the limitations of the pre-existing 35°C wet-bulb temperature model.

In general, the paper needs to talk more about smooth gradients, for example the idea that the 65 age cutoff you use is not a perfect binary. What about people with health conditions? What about people with different constraints/goals in their activities (Horton et al. 2021)? Adaptation options like putting cold water on yourself? Note that I am not suggesting new analyses here. Rather that there is a chasm between the needs you articulate in the introduction, and the nice additions you address through this nice study. In the discussion, circle back to remind us how your study helps fill in some of the missing pieces, but a chasm remains before anyone is able to fully model/understand all the factors that influence the relationship between humid heat and mortality/liveability.

Response: Thank you for this. We note that it's unclear what we address versus what our new approach will allow us or others to address in the future. Some of these ideas you mentioned are exactly the type of future work that could be done. However, as the first paper that demonstrates the entire model to a new audience, as well as the proof-of-concept in climate models, we were careful not to overload it with nuance. With respect to ageing, it is true that a person will not suddenly have problems sweating on their 65th birthday, however it has been well demonstrated in the physiological literature that the effects of primary ageing on thermoregulatory capacity (primarily mediated through changes to sweating) are not linear, and that marked decrements tend to emerge between the ages of 60 to 65 y. It is certainly true though that progressive aging effects are observed with further ageing and the current model does

not capture this. However, by basing our “older” age group on parameters that most closely resemble ~65 y, we are avoiding any potential exaggeration of the primary ageing effect on heat vulnerability, which we demonstrate to be quite pronounced even with our more conservative approach.

To address this concern, we think that **Figure 1**, the new **Table 3**, and the final paragraph added in the introduction are vital. Also, in the introduction, we now have added “As an initial step” when we introduce exactly what we do “i.e., *As an initial step towards these advances, we apply a whole-body human heat exchange model to estimate heatstroke deaths and maximum/safe activity across **two subpopulation types (younger (~18-40 y) and older (>65 y) adults)**, considering **sweat rate impairments** (due to older age) and heat exposures in the **sun or shade** and across an array of air temperatures (T_{air}) and humidity levels.*”

Second, at the end of the discussion, we have added:

Lines ~389-392: “*These analyses and discussion highlight the capabilities of our approach, and within the Supplemental Material (SM) we note the capabilities and current constraints of the model. **Table 3** indicates future work and capabilities of the model that will support addressing further gaps and shortcomings discussed in this study but not examined quantitatively, requiring significant collaboration among disciplines.*”⁷¹”

Finally, we have made it very clear in the conclusion precisely what gaps we fill and what may remain, as follows:

Lines 397–401: “*This paper establishes a new physiological modeling approach for assessing human survivability and liveability across subpopulations (younger and older adults) and in sun or shade, designed to be suitable for application with global climate model output. Thus, we fill specific gaps around these factors across the full spectrum of temperature and humidity combinations, and introduce a brand-new assessment to determine liveability.*”

Table 3: Future work to be accomplished and tests to run with the newly presented modeling approach to allow for numerous analysis types and answers to larger questions.

Sub-Topic or Discipline	Future Work/Next Steps based on model capabilities
Thermal Physiology	Create a catalogue of specific estimates of skin wettedness, skin temperature, and maximum sweat rates across population types, ages, medications intake, and health disorders/chronic diseases.
	Modeling the impact of personal cooling strategies, such as dousing or misting skin, electric fans, and foot or hand immersion in water. ^{79,80}
	Conduct a sensitivity analysis of wind flow impacts within the liveability and survivability models, and activity velocity impacts for liveability.
	Model changes behavior based on thermal exposure (e.g., Vargas et al. ⁸¹) and other physiological attributes that affect adaptive behavior.
	Characterize other measures of physiological heat strain, including cardiovascular and renal strain (not only heat stroke, as modeled here).
	Test and model different clothing factors within the liveability and survivability models.
	Empirically test maximum duration of intensity safe activities (M_{max}) in different extreme environmental conditions and describe the changes in skin temperature and sweat rate over

	time and across the lifespan (e.g., ^{40,63,82}) to allow for appropriate ranges and scaling factors applied within both liveability and survivability models.
	Improve the steady-state version of the model, as applied here, to account for changes within the 3–6H exposure in the variables assumed as constant.
	Conduct experiments across a range of plausible biophysical parameters for uncertainty estimation and subsequent model application across different temperature/humidity combinations along the survivability and liveability threshold curves.
Climate Sciences	Run the present model using multiple GCMs and emissions scenarios to provide an ensemble of futures, addressing various sources of uncertainty, including that from both the biophysical and climate models (e.g., Petkova et al. ⁸³).
	Obtaining regional or city-specific analysis of future heat stress with downscaled, bias-corrected climate projections.
	Evaluate time-of-emergence for different survivability and livability thresholds using single (global climate) model initial-condition large ensembles (SMILEs).
	Assessing the relative importance of human population trends (increase in the number of people, aging) versus climate change in future health burdens from extreme heat.
	Quantifying the probability of exceptional heat wave and/or mass heat fatality events over different regions with different levels of climate change.
Public Health	Determine the sources of differences between physiological models and epidemiological models to inform future model improvements.
	Quantify compounding and cascading risks connected to heat and health, including heatwaves compounded by wildfires.
	Evaluate the extent to which different urban designs would alter liveability or survivability.
	Leverage the new modeling capabilities along with projected global demographics to better determine impact across a diverse population for public health preparedness.

Most fundamentally, you need to be much clearer about which aspects of the gaps in our understanding/prior work your analysis addresses, and which ones it doesn't. For example, lines 49-52 read almost as legerdemain, that needs to be cleaned up. I think you can still make your overall point, even if it means adding a sentence or two, in a way that acknowledges that some epidemiological/econometric approaches (even with projections) have included humidity. See my comments above about how you need to engage epidemiology/econometrics/empirical approaches a bit more. This is one of the major branches of analysis in assessing heat impacts on health, so it needs significantly more attention here. This can help strengthen the case that a reader should instead trust your single physiological model result!

Response: Thank you for this feedback, we definitely want to make sure that these aspects are very clear! We think the responses to the multiple comments above will help ensure it is clearly stated what gaps we do and do not fill. Regarding the examples from Lines 49-52 in the original draft, we have revised that paragraph and the following to be more balanced regarding the benefits and limitations of epidemiology/econometric and physiology-based studies of heat-health outcomes. We also now are clearer that several epidemiological studies have considered humidity and include example citations to such studies, even though the current leading large-scale study finds a negligible role for humidity in heat-health outcomes. While the scope of this particular paper focuses on a new physiology-based method, we fully agree that both approaches (epidemiology/econometric vs physiology) play important roles in understanding the future impacts of heat. We also think it's important to note that survivability models predict widespread death. Using historical epidemiological data to achieve that is not really relevant because (thankfully) it has not happened yet.

We hope our revisions below successfully provide this more balanced perspective.

Paragraphs on lines 51–58 now reads as:

“Empirical approaches are based on real-life outcomes and the range of realistic living conditions, and they can explore cumulative effects of exposures over multiple days. However, two limitations for climate change projections include 1) assumptions needed to extrapolate results to warmer temperatures than observed in the historical sample⁸ and 2) ambiguity regarding the role of humidity in heat-health outcomes.¹⁹ While some epidemiological studies find a relationship between mortality in the heat and humidity¹¹, most find minimal associations between humidity and heat-health outcomes¹². Given that specific humidity is robustly expected to increase with global warming, this is a key research gap for epidemiology-based projections of future heat stress.”

Lines ~60-66:

“Physiology-based studies of future heat stress risk employ relationships between the thermal environment and health outcomes based on human energy balance considerations, with parameters constrained by studies of physiologic processes. In contrast to epidemiology/econometric approaches, physiology-based studies of heat-health outcomes consistently find a robust role for atmospheric humidity in heat stress via its modulation of cooling from evaporation of sweat¹⁰. However, physiology studies are limited in not directly observing extreme health outcomes, such as hospitalization or death, and employing idealized conditions from thermal chamber studies. A range of physiology-based metrics have been employed to project future heat stress.”

I think the Introduction is generally strong—well written, especially when describing gaps and limitations in prior work. That said, I needed a little more info before the paragraph on lines 88-92. Despite excellent text before this paragraph, the 43C threshold still comes seemingly out of nowhere in the first sentence. And then in the second sentence we hear about maintaining core temperature, which leads one to wonder about the gray area between these two extremes. I don't question the use of these two measures—we just need some added context before you introduce these two specific 'lines in the sand' after talking too us earlier about all the nuances that can render lines in the sand blurry (pre-existing health conditions, adaptations, etc.).

Response: Thank you for this feedback. We have tried to ensure this is not such a large jump between the two paragraphs by having more flow when discussing heat stroke death at a T_{core} of 43°C. We also ensure that it is clear that 43°C is a core temperature value, which is what we use in our model as heat stroke death.

The following has been added:

Lines 106-109: *“A human would experience heat stroke death from hyperthermia on 99.9% of occasions when an individual's core temperature (T_{core}) exceeds 43°C³² (see **Table 1**). Thus, we define the limit of **survivability** as reaching T_{core} of 43°C in 3- or 6-hour exposure windows to allow for comparison with the T_w of 35°C assumption (heat stroke death after 6 hours).”*

Lines ~112–116: *“The realistic final T_w value for the limits of survivability or liveability will differ by person and climate type (dry versus humid; sun versus shade), and thus is flexible (i.e., the limit will differ). Hence, while we state a final T_{core} at which heat stroke death will almost inevitably occur, our*

approach does not assume a unique T_w threshold; rather, hundreds of T_w thresholds are possible depending on differences in people and conditions modeled, with wide-ranging opportunities of the model in future research (Table 3)."

Again, your summary of what is needed, and the importance of more nuance in these studies is strong. It is just odd that you aren't more precise about which of these many pieces you all chew off in your analysis. Without a close read, one could easily assume your study addresses ALL the shortcomings of prior studies you have just (nicely) elucidated, rather than just a couple (important ones).

Response: Thank you for this suggestion. To address this, we added Table 3 to show future work, improved the framework (now Figure 1) to show what we do in this study clearly, and added the below paragraph to the end of the introduction:

*Lines 130-135: "An overview of environmental conditions, populations modeled, and assessment types examined in the present study is provided in **Fig.1**, which displays the scenarios and tests this study accomplishes that were disregarded by other studies. We do not demonstrate the sensitivity of illness or health status (including acclimatization) in the population, activity velocity, personal cooling strategies, body shape and size, or an ensemble of climate projections, all of which are needed in future work (**Table 3**)."*

The following is already stated after the goal in the introduction, on lines 121-123:

"We focus on two subpopulation types (younger (~18–40 y) and older (>65 y) female adults), considering sweat rate impairments (due to older age) and heat exposures in the sun or shade and across an array of air temperatures (T_{air}) and humidity levels."

Along similar lines, the critiques of prior work on lines 66-67 seem fair, but seem to promise more than you deliver. For example, in contrast to prior authors, do you consider body size, or more than a single (age 65+) restriction to thermoregulation? Why not say that you don't?

Response: Indeed, we should say what we do explicitly and do not test in this study and what the model capabilities are moving forward. As stated in a response above, now we provide the information of what we DO versus what we DON'T address based on the gaps presented, stating that we take the "initial steps" needed in this inaugural research paper to start to tackle this enormous interdisciplinary challenge.

The following is added to the introduction:

*"An overview of environmental conditions, populations modeled, and assessment types examined in the present study is provided in **Fig.1**, which displays the various scenarios and tests this initial study accomplishes that are disregarded by other studies. Specifically, we address sun versus shade, compare age (young versus old), allow for clothing and metabolic rate considerations (in the liveability analysis), and apply to two climate projections. We do not yet demonstrate the sensitivity of illness or health status (including acclimatization) in the population, activity velocity, personal cooling strategies, body shape and size, or an ensemble of climate projections, all of which are future work needed in the community (**Table 3**)."*

We also believe that the new **Table 3** will delineate what still can be done with this work and that Figure 1 shows better what IS accomplished in the current study.

3. I am trying to figure out how in the extremes of high T and low RH the downward revisions of the danger thresholds relate/don't relate to actual weather conditions (as opposed to theoretical or lab experiment-driven conditions). Once temps get really high (e.g. 115+ F) on your figures, there currently aren't any RHs of 40 percent, for example. Along similar lines, does any literature suggest that anything like 35C wet bulbs occur or will occur at low RH? In a sense my point is that not only does this paper focus on a single physiological/human model, not only does it largely punt on describing humid heat epidemiology, but it also largely punts on describing the climatology of observed heat and humidity. It is critical to know which combinations of T and RH are experienced and likely to be experienced. As a bare minimum revision, can we see reported combinations of T and RH somewhere, perhaps added to some of the existing figures 1-4?

Response: The graphs generally show combinations of temperature and humidity that are indeed reached in many areas of the globe. SM Figure S14 and S15 shows 3-H data for 2016-2025 across selected hot-humid and hot-dry locations, demonstrating that extremely hot and dry and extremely hot and humid combinations can be reached; those conditions will also become more likely in the future, and thus we need to be able to model the physiological impacts. Reviewing current literature on heat extremes, we note that the occurrence of T_w above 38°C is not very plausible, and thus remove any isotherms above that level. That change has been done for all graphs. The election of T_w 38°C is based on the report of some very limited T_w 35°C occurrences already (Raymond et al 2020) and assuming that 3°C would be a reasonable threshold to encompass conditions reached in hot and humid tropical locations with future warming, given work showing that extremes of wet bulb temperature in the tropics scale closely with tropical mean tropospheric temperatures (Zhang et al 2021). Finally, we must remember that although some conditions may not be reached much outdoors, they could be reached indoors in factories, mines, etc.

In the discussion, we now refer to the SM figure that shows these conditions, as well as state the following with two new citations:

“Results encompass current and future extreme heat (see SM Figure S14-S15 for current ranges) across very hot and dry and very hot and humid situations with risk increasing, or expected to increase, in parts of the world.”^{32,34}

We also have added the following sentence in the results:

“Results are shown for moderate to very extreme combinations of temperature and humidity that may be reached at times today, both indoors and outdoors, but more frequently in the future.”³²

Also, in the supplemental material, we mask out areas of the water vapor and specific humidity graphs with RH >100% (non-plausible conditions).

We do not feel we need to describe the climatology or epidemiology of heat and humidity when many other papers have done so, many of which we refer to in how we justify the need for this work as well as discuss the connections to other literature and global impacts. Doing as such would deter us from our goal, and we also do not have the space. Also, with respect to humidity, it has not been identified as a

strong predictor of heat-health outcomes to date (see Baldwin et al., 2023), possibly because those who do die in from heat are very physiologically vulnerable and perhaps have blunted sweating ability.

Finally, with respect to the model, it is indeed one physiological model with numerous inputs. It can be thought of like a weather or climate model with many input variables or starting points. With humans, there's not one human*, but there is one set of fundamental principles (like climate models) to follow. So we can model various types of humans with our one model by using different variables for age, sex, body size, clothing, sweat rate, acclimatization, etc. etc. In the end, we will be able to make "ensembles" of many types of humans with the one model.

*Note that the Tw of 35oC approach does assume there's only one type of human all of same body size, sex, etc.

Citations we have used and added:

Christidis, N., Mitchell, D. & Stott, P.A. Rapidly increasing likelihood of exceeding 50 °C in parts of the Mediterranean and the Middle East due to human influence. *npj Clim Atmos Sci* **6**, 45 (2023).

Raymond, C., Matthews, T., & Horton, R. M. (2020). The emergence of heat and humidity too severe for human tolerance. *Science Advances*, *6*(19), eaaw1838.

Zhang, Y., Held, I. & Fueglistaler, S. Projections of tropical heat stress constrained by atmospheric dynamics. *Nat. Geosci.* **14**, 133–137 (2021). <https://doi.org/10.1038/s41561-021-00695-3>

4. One GCM, one RCP is a problematic. For Nature Communications, I think you need to include more models in the main, and probably another RCP in the Supplement. Note that these additions would not starve the follow up study you refer to about sources of uncertainty, in a field with so many possible permutations (Petkova et al. 2014, for example.)

Response: Thank you for this comment. Based on this and a related comment from reviewer 2, we have added another RCP and another model in the paper, and thus now have assessed liveability using the following models and scenarios from CMIP6: GFDL ESM4 and MPI ESM1.2 following SSP2-4.5 and SSP5-8.5. These revisions affect Figures 6 and 7 in the main manuscript (including captions), manuscript text in Section 4.4, and SM Figures S9-S13. The results from both models are qualitatively similar for both the global maps and assessments of particular locations. However, as expected, there are greater decreases in liveability for SSP5-8.5, related to the steeper increases in carbon emissions over the next century. We agree that a more extensive suite of GCMs and scenarios would be useful to explore in future work (as mentioned in Table 3 above). Still, we hope that this present analysis serves as a valuable proof-of-concept for this initial paper, focused predominately on describing and demonstrating the capabilities of the physiological model.

As part of this change, the methods have also been updated.

Line by line comments

Liability' in title to 'Liveability'

Response: Thank you for catching this. We have also changed the spelling from livability to liveability, which is the more preferred version.

Abstract, second sentence: move the last 9 words (perhaps further shortened) up to earlier in the sentence. As written, it seems like they only apply to livability not survivability.

Response: We have moved the “in sun or shade” portion up, but we only do the projected climates for liveability, so leave that with #2.

Introduction

Is the term ‘heatstroke death’ sufficient here? (I am wondering if there are other kinds of heat related deaths you mean to include that are not captured by the term ‘heatstroke death’)?

Response: Yes, it is sufficient in this case because the methods used here for survivability and those using the Tw of 35oC assume a rise in core temperature to critical limits that would cause a heat stroke death. While cardiovascular deaths and kidney failure are two other common causes of heat death, we are not modeling those in the survivability portion of this study. These differences between the types of deaths are stated in the introduction starting on line 100 where we discuss this and state:

“This study and related studies assessing survivability limits (e.g.,^{12,14,16}) estimate deaths arising from critical high core temperatures. A complex cascade of events^{30,31} ultimately leads to multi-organ failure^{3,310} and often death³¹. Hence, we model heat stroke deaths (hyperthermia) and do not model the two other common types of heat-related deaths: death from cardiovascular collapse and renal failure and collapse, acknowledging that heat stroke deaths are a fraction of total excess heat-related deaths.”

Further, in the supplemental material we state:

“This paper and others using limits to adaptability or survivability frameworks focus on heat stroke deaths (hyperthermia) and do not model the two other common types of heat-related deaths: death from cardiovascular collapse and renal failure and collapse, acknowledging that heat stroke deaths are a fraction of total excess heat-related deaths.”

To make it more evident in the main paper, we have added the following sentence:

Line ~102-104: “Hence, we model heat stroke deaths (hyperthermia) and do not model the two other common types of heat-related deaths: death from cardiovascular collapse and renal failure and collapse.”

This sentence also allows for better flow into discussing the critical thermal maximum of 43°C, which was a comment above.

Penultimate paragraph—clarify that you mean an individuals aging, rather than the average age of humans (or more to the point the number and age of older people) going up in the future

Response: Thank you. We have clarified that we mean older age.

‘3.2’ at first use in heading to 2.2’

Response: Thank you. Fixed.

‘3.3 at first use in a heading to 2.3’

Response: Thank you. Fixed.

I feel Figures 1 through 4, while all excellent, are so similar stylistically and content-wise similar as to

argue for making a hard decision to move at least one of them (possibly two of them) to the supplement.

Response: We feel these are all important and show very different things, so we have left these four, but to reduce the number of figures, we have moved Figure 8 to the supplemental material.

Line 273: 'Plan' to 'Plain'

Response: Thank you for catching that. Fixed.

Figure 3: Give us a better sense, through examples in the text of what levels of activity different values of 'Met' correspond to. Figure 3 is nice in this regard, but say more in the main text.

Response: This is a great idea. We have updated the text in the results to give more context when referring to specific MET levels, and have added some more specific examples in the caption of Figure 3.

Also do the icons, like running, assume no rest during 3 hours?

Response: Correct, it implies a continuously sustained intensity in the activities while maintaining a core temperature at a steady state. There is no attached duration for the safe activity in the primary liveability analysis, just an intensity. For determining areas of survivability but NOT liveability (hence full rest)—shown by the hatched areas—we needed to include a duration component. Therefore, the 3H duration in the graph is only to display the exposures of people during rest and storing heat due to uncompensable heat stress, when the survivability limit for 3H of heat storage is not reached. For areas to the left of the hatched areas, the representative duration of the activity to remain compensable will depend on steady sweating conditions, skin temperature and thermal environment. If a person's activity level greatly exceeds M_{max} , the body will be in a state of continuous heat storage (uncompensable heat stress), and in danger; then, they need to slow down the activity or move to a colder environment. We have ensured that this continuous effort and steady state of environmental and corporal assumptions are explicit in the methods and caption of Figure 4.

Figure 5a: why are some areas showing reduced MET?

Response: Many areas show reduced MET because we are demonstrating how much LESS work people can sustain safely (without a continuing rise in T_{core}) as the conditions will change in these locations between today (contemporary) and end-of-century, assuming no additional forms of adaptation to heat are used. Therefore, due to heating, many places will have conditions where only reduced work ability is possible. Based on this comment and the next, we have updated the Figure 6 caption as follows to avoid confusion and make sure it doesn't appear counterintuitive. We have also changed the legend in the maps so that the largest M_{max} decreases are displayed as the largest negative values at the lowest fraction of the color palette.

We added the following to Figure 6 caption:

"Activities by MET level range from no activity (sitting ~1.5 METs), to housework (~3.0 METs), dancing (~5.0 METs), and heavy lifting (~7.0 METs)."

Figure 5b: worth exploring why Bangladesh has the lowest values. I think of this area as closer to high RH low dry bulb than many of the other humid heat hotspots around the world like the Indus Valley for example. Given your findings (high dry bulb at low RH is more dangerous than most researchers have realized) this is somewhat counterintuitive. Can you look into some explanations? (Maybe GCM resolution, maybe the sheer number of days per year is a more dominant term in your model [maybe Bangladesh has a lot of pretty oppressive days, rather than having the individual days that are the most extreme], etc.).

Response: Bangladesh has some of the highest reductions in metabolic activity due to sustained hot conditions, meaning that the conditions are very oppressive because of magnitude and frequency, as you pointed out. In the supplemental material, Figure S12 shows the percent of time with $T > 25^{\circ}\text{C}$. In Bangladesh, in the current decade it's around 50% and will increase (to ~60-70% depending on the SSP and scenario). Also, focusing only on that warm time, the relative frequencies in the 2D histograms (Figure 7) shows a predominance of very humid conditions (above 80% RH between 28 and 32C). This is why in Figure 6b and e, results show that people can do less in the decade average. So the colors over Bangladesh show that it is actually one of the worst locations when it comes to heat and humidity because the maximum activity that can be done at the of the century is very low and for a high proportion of time. We have to clarified this in the caption so that it doesn't come off as counterintuitive.

Figure 6 caption: 'Dix' to 'Six'

Response: Thank you for catching that. Fixed.

Figure 7: I think this should be much earlier in the paper.

Response: Thank you for this suggestion. We were thinking it should go with the methods, but since the methods come after the results and discussion in this journal, we have found a way to make it appear just after the introduction.

Figure 8 caption: Is 'wittedness' intended? Especially if you are constrained in your number of Figures, this is a very logical figure to move to the Supplement.

Response: Thank you for catching that. This should not be "wittedness". We have fixed it to say wettedness. Also, thank you for this suggestion about the figure. We have moved it to supplemental material, where we detail the full model.

Methods 4.4. did you bias correct for cities? If not, it is OK, but you should say so.

Response: We did not for this particular paper. This has been added in section 4.4 to the methods.

References

Horton, R. M., de Sherbinin, A., Wrathall, D., & Oppenheimer, M. (2021). Assessing human habitability and migration. *Science*, 372(6548), 1279-1283.

Petkova, E. P., Vink, J. K., Horton, R. M., Gasparrini, A., Bader, D. A., Francis, J. D., & Kinney, P. L. (2016). Towards more comprehensive projections of urban heat-related mortality: estimates for New York City under multiple population, adaptation, and climate scenarios. *Environmental health perspectives*.

Reviewer #2 (Remarks to the Author):

The manuscript focuses on assessing survivability and livability by applying physiological and biophysical principles. The authors suggest that the 35°C T_w threshold underestimates the risks for older adults. They also suggest that the risk continues to increase in a changing climate.

I have two major concerns with the manuscript. First, the authors use only a single GCM and a single scenario that fails to capture the range of future climate uncertainties. Secondly, the authors extract city-level data from a pretty coarse resolution GCM. The majority of GCMs in CMIP6 are still pretty coarse

resolution including the GFDL-ESM4 model that the authors have used. Using GCM data to draw such conclusions can be misleading given that it fails to capture various city-specific features and will have substantial biases. From the title of the manuscript, it looks like the authors are trying to evaluate the survivability and livability in a changing climate, and therefore, using single coarse resolution GCM is not justified. A minor comment- it should be “livability” and not “liability” in the title.

Response:

First, thank you for catching the error in the title. We have also changed the spelling from livability to liveability, which is the more preferred version.

Second, the example with the GCM is intended as a proof-of-concept or case study to demonstrate the capabilities of the physiological model, but we see that there is a need to show more than just one model and scenario with the new method to build confidence. Following your comment and one by another reviewer, we have added another RCP and another model in the paper, and thus now have assessed liveability using the following models and scenarios from CMIP6: GFDL ESM4 and MPI ESM1.2 following SSP2-4.5 and SSP5-8.5. These revisions affect Figures 6 and 7 in the main manuscript, manuscript text in Section 4.4, and SI Figures S9–S13. The results from both GCMs are qualitatively similar for both the global maps and assessments of particular locations. However, as expected, there are greater decreases in liveability for SSP5-8.5, which have steeper increases in the carbon emissions over the next century. We acknowledge that a more extensive suite of GCMs and scenarios would be useful to explore in future work, including a more comprehensive climate change uncertainty analysis (as mentioned in Table 3 above), but hope that this present analysis serves as a valuable proof-of-concept for this initial paper focused predominately on describing and demonstrating capabilities of the physiological model.

Thanks also for noting the significant limitations of GCMs when looking at particular cities. We agree that ascertaining results for specific cities can be tricky with relatively coarse GCM output and that ideally, such data would be bias-corrected prior to analysis. However, we still think there is a utility to contextualizing the effects of aging against the distribution of temperatures for particular locations (as in Figure 7). Given this, we retain that analysis, but have revised Climate Analysis methods text (line 503) to say we explore the distribution of liveability in present and projected future conditions in particular regions across the globe, as opposed to specific cities. Additionally, the addition of the second GCM, which has similar results for this analysis, adds some confidence regarding the robustness of the results. We also acknowledge that future work should conduct similar analyses for particular cities using downscaled, bias-corrected projections (as mentioned in Table 3 above). Please see relevant text revisions in lines 533 to 535. We hope this provides a middle ground that adequately addresses the reviewers’ comments.

Reviewer #3 (Remarks to the Author):

OVERALL:

The goal of the paper by Vanos et al. was to improve upon the modeling of survivability and livability limits of humans exposed to increasingly hot environments due to climate change. In many ways the authors succeed in their goal of advancing upon earlier modeling efforts. The authors, in particular, extend survivability and livability limits to very hot and very dry environments and attempt to show differences between young and old. Other advantages are mentioned (contingencies for disease states, acclimatization status, body size, fitness, etc.) but not described. The work submitted is in many ways masterful, but there are numerous potential problems with the many assumptions made and as a result

the **interpretations also appear problematic – even incorrect**. A model such as the UTCI, validated against empirical data (such as Brode and Kampmann, 2023), may be a better approach? Questions and comments follow for consideration.

Response: Thank you for these overarching and helpful comments.

First, we indeed state the potential of our approach to address aspects like body size, fitness, acclimatization status, and disease states but do not specifically test these in the given paper. Thus, we have ensured it is clearer what we demonstrate in the current paper (age group differences, sun versus shade, sweat rates, use with climate projections) versus what the model has the ability to do in the future based on how the new methodology is set up. We also direct you to Reviewer 1's comment #2 above with a similar request.

Second, for the UTCI, please refer to response #5 below. Briefly, the available regressions from the UTCI do not allow for studies motivated to be made for different types of people (e.g., age, standard body size, clothing assumptions, metabolic rate, walking speed) and thus are unsuitable for the given goals of this work.

Finally, we hope that our responses below and to the two other reviewers help provide more understandable interpretations of the data so that they do not seem problematic. We notice that directing this article to various types of disciplines, while important, could result in some misunderstandings of terms, purpose, and assumptions within the paper, and we have attempted as best as possible throughout the paper to avoid such instances.

MINOR:

Line 1: Title - liability should read livability.

Response:

Thank you for catching this. We have also changed the spelling from livability to liveability, which is the more preferred version.

Lines 183, 191: A decrease in Mmax of 0.2-0.26 METS does not seem like an appreciable difference (10-20% can be the measurement noise for METS). Can the authors better contextualize how this matters to human health, economic burden, etc.?

Response: This is a nice idea. Note that the global median of the Mmax decreases (GFDL model) for SSP 2-4.5 is -0.25 METs, and for SSP 5-8.5 is -0.64 METs. Both results are representative of conditions >25°C. We have added the following for context:

Lines ~346: "These changes could represent a slightly lowered productivity (e.g., less crops harvested, the need for extra workers) or decreased activity performed, with economic or health consequences.⁵⁸⁻⁶⁰ While quantifying these consequences is beyond the scope of the present paper, this should be the focus of future research."

Line 372: Did you mean reference #30 for this statement? Reference #67 I understand.

Response: Thank you for catching that. We mean Cramer and Jay (2019). This is fixed.

Lines 389-403: It would be helpful for those interested to introduce the average height and weight used

to arrive at the 1.60 m² (young) and 1.78 m² (old) (supplementary information) values that appear to have been used for the calculations represented in all the tables and graphs. Were these population median values for height and weight in young and old populations?

Response:

Indeed, these were population mean values for young healthy women and older women. We used the average for women since studies rarely look at women (they always use the averages for men). The model can be used for any height and weight.

We have added the following info and citations to the SM (references are based on the SM reference list).

“These surface areas arrive from Eq. 7 and are based on global average height for women⁸ (~1.62m) and average weight data based on ExposFacts^{9,10} and the EPA Exposure Handbook¹¹ (using NHANES data) to ensure a more representative global average, resulting in ~57kg for young female and ~74kg for older female.”

Note we also have made sure it clearer throughout all of the text that the values are for young women, not overall adults.

Line 406: It does not seem reasonable to use a nude body for survivability (how many people would be nude?) but minimal clothing for livability. I understand wanting to compare your results to previous work, but the buck should stop somewhere? Or consider using nude estimates for livability as well?

Response: We use a conservative approach. This is the same assumption (i.e., a person being nude) by using the T_w of 35°C thresholds, so we have them nude to make a direct comparison. It is our position that if a person is suffering from excessive heat stress to the extent that their survival is threatened, removing clothing to allow for more heat loss would be a plausible adaptive behaviour. In terms of the clothing selected for our liveability analysis, it is our position that for an environment to be considered “liveable” in most parts of the world, estimations of heat transfer must account for at least some clothing – otherwise our estimates would define conditions that are only “liveable” if everyone is nude, which does not seem especially useful. Please also see the new Table 3, which provides other future work where we can test clothing changes with the model.

References: There are a number of references with errors in them. Please review and edit.

Response: Thank you for noticing this. We have reviewed and updated references for accuracy throughout.

Table 2: The lower T_w limit at lower humidity seems strange without the context of the very high accompanying air temperatures.

Response: This is a great point. We have added the corresponding air temperature to Table 2 and the Supplemental Table. Also, note that the horizontal lines in Figure 3 help readers see the T_{air} value at the given RH to result in the final T_w limit. We have also ensured that the section describing this table is updated with corresponding air temperature values.

This addition is now provided in Table 2 caption (“...along with corresponding air temperature. Each humidity level is also by the horizontal lines on Figure 3.”) and we have made this fact clearer within the

Figure 3 caption (“...horizontal dark blue lines indicate the ΔT_w values at RH levels of 10, 25, 50, 75, and 90, and 100%.”)

Figures 1-4: Where are environmental conditions projected to rise to 50 degrees C or more to make these projections less theoretical and more meaningful?

Response: We have provided this context in the results for examples, but there are many regions where temperatures are or have already risen (and are projected to rise) to 50 degrees or more. So we extend the graph this far to be able to quantify those conditions. For example, an article by Christidis et al. (2023) shows a rapidly increasing likelihood of exceeding 50 °C in parts of the Mediterranean and the Middle East in the future, and thus we need to be modeling these conditions. We have cited this paper in the study and stated our reasoning for the ranges we have selected to ensure we are reaching conditions that the future may hold for some locations.

Christidis, N., Mitchell, D. & Stott, P.A. Rapidly increasing likelihood of exceeding 50 °C in parts of the Mediterranean and the Middle East due to human influence. *npj Clim Atmos Sci* 6, 45 (2023). <https://doi.org/10.1038/s41612-023-00377-4>

Figure 6: Can the authors explain the y-axis probability?

Response: Thanks for catching this. The y-axis in the histogram refers to the relative frequency of the M_{max} values for each histogram bin (i.e., percent of the total values that fall in each bar range). This is already addressed in the graph, and the new label says “Relative frequency (%)” not probability.

Supplemental Material and Model Assumptions:

Equation 20: How is S_{max} determined? Is it simply E_{req} ?

Response: Thank you for catching that. We have added the values for S_{max} and the studies that determined them. The values are: 0.75L/hr (18-40 years) and 0.51 L/hr (65+ years). These have been added, as follows, after equation 20:

“where S_{max} is $0.75L\ hr^{-1}$ (18–40 years) and $0.51\ L\ hr^{-1}$ (65+ years)¹⁸...” (Morris et al., 2019).

We are being quite conservative about the prospect of survivability with the sweat rates we have chosen, as there is such a variation in the population. Higher sweat rates would shift our line a bit closer to the T_w of 35°C line. Also note that we are also saying that these sweat rates are sustained for up to 6 hours without any dehydration effect, so we do not want to be overestimating sustained sweat rate.

MAJOR:

Line 389: The premise for using 43 degrees C for survivability is justified by citing the range (41-47), but 40.8 degrees C was the average exertional heat stroke temperature observed in more than 100 military cases (see Figure 5-3 in:

https://armypubs.army.mil/ProductMaps/PubForm/Details.aspx?PUB_ID=1024722). Is 43 degrees C more applicable for passive heat stroke? Should a distinction be made between passive and exertional heat stroke? Which are you modeling, specifically? Both? A more justified value ~41 degrees C could drastically change your results and interpretations.

Response: We are modeling classic/passive heat stroke, not exertional, as in the survivability model people are at rest. The 43°C is quite a well-established upper limit (thermal maximum) in the literature. Indeed, it may be much lower (such as the example that the reviewer helpfully cites), yet there is a large range among people, and often with passive heat stroke cases, the actual T_{core} reached is unknown. Thus, we are aware our 43°C is very conservative (i.e., it is likely that most people will suffer heatstroke below this value), which may offset any concerns about us not being liberal enough with sweat rates. As an important note, the Sherwood and Huber 35°C T_w model would result for a 6-H exposure in a T_{core} of 48-50°C depending on body size. Yet as that is the model used across most studies in this space currently, we hope this work provides a large incremental improvement upon these assessments. Moreover, as stated in our methods already, “the translation from heat storage to T_{core} between our physiological approach and the 35°C T_w limit is not the primary driver of their differences; if it were, the “3-h shaded” model (dotted blue line, **Fig.2a**) would overlap the 35°C T_w (thick solid black) line. Instead, substantial differences persist, especially above T_{air} of ~45°C.”

Reference: Bouchama, A. *et al.* Classic and exertional heat-stroke. *Nat Rev Dis Primers* **8**, 1–23 (2022).

Lines 432-436: I am not sure I understand $S = 0$ in this explanation. Heat storage could and would take place to elevate body temperature (from 37 degrees C) to a new but stable baseline (e.g., 37.5, 38.0, 38.5,...). Why must no heat storage be assumed (not realistic)?

Response: We regret we were not clearer in our original submission. We are not saying cumulative (absolute over time) heat storage is 0, but that a rate of heat storage of ~0 can be attained. While there could be an elevated steady-state core temperature after activity commences, once thermoregulatory responses are engaged, core temperature will reach a steady-state plateau. We have improved the clarity on this in section 4.3.

Supplemental Material and Model Assumptions:

1. Is airflow 1 m/s used in both survivability and livability scenarios? In Figure 3, a MET value of 2.5 (walking) arguably creates airflow around a person equal to ~1 m/s, but jogging at 7.5 METS would be more like ~2.25 m/s. Riding a bicycle at 5 METS would produce an airflow of ~5 m/s, completely ablating the insulative air boundary layer. The under-estimation of airflow will have important effects on your interpretations that rely so heavily on evaporative cooling.

Response: We use 1m/s wind speed for both. One reason is to match better with Sherwood and Huber. For survivability, the person sedentary, so self-generated air velocity is <0.2 m/s. This would always be the case for someone laying down and allows our results to be comparable to the T_w of 35°C approach. The airflow is held constant to differentiate between the impacts of the temperature, humidity, and solar conditions on the survivability line. For the liveability analysis, we also use the 1m/s and do not alter activity speed either, and we agree more robust incorporation of activity speed would be a useful area for future work (as noted in the new Table 3), but is beyond the scope of the goals of this paper. We’ve made sure these points are clear in the SM.

2. Mean T_{sk} is set to 35 degrees C. Though it may align with other publications, mean T_{sk} can be easily 36 or 37 degrees C in very hot air (40 to 50 degrees C), especially when globe temperature is higher still. Some military models use 36 degrees C or establish a scaling factor for the best gradient. This too will potentially impact your interpretations and calculus as dry heat gain is somewhat exaggerated.

Response: We acknowledge the skin temperature can get higher, yet we have chosen this value for this initial study based on the literature consensus that mean vasodilated Tsk of 35°C, but also the Tw of 35°C wet bulb threshold model by Sherwood and Huber uses a Tsk of 35°C assumption as well. Hence, it allows us to compare this approach, which is one of the main goals. These reasons, with citations, are already provided in the SM modeling information.

Nevertheless, we certainly acknowledge that this is limitation and we have included this as a major area of future research and application using ranges and scaling factors in the new Table 3.

3. The skin wettedness values employed of 0.50 to 0.65 seem too low, especially 0.50 (e.g., Candas et al., 1979; Ravanelli et al., 2018). This makes a huge potential difference to Emaxlim. Can the authors provide justification for such small values? If 0.65 to 0.85 were used, how would this impact your results? I suspect greatly.

Response: We initially were trying not to overestimate skin wettedness so we used lower values but in reviewing the literature, we agree that 0.65 and 0.85 are more appropriate based on Candas 1979 and Morris 2021. We have now changed these values in the analyses and repeated all modeling runs.

We have made these updates to the detailed methods in the supplemental material while also recognizing in Table 3 that future work should vary the sweat loss between subpopulations to account for a range of cooling potentials.

4. There is a major emphasis on how age impacts the results of the model, but little explanation is given for why. Reference #15 is cited for age negatively impacting Emaxwet. However, while reductions in vasomotor and sudomotor function have been convincingly demonstrated to occur with aging, translation to heat balance on the whole body level is less clear. For example, men 45 years older than younger subjects matched for body mass, height, and body surface area had 50% lower local sweating rates, even though Ereq in W/m² was similar (Schmidt et al., 2022). Others have quantified an age-related reduction in thermoeffector heat loss potential of ~4% per decade between ages 18 and 70 (D'Souza et al., 2020). How does this compare with your model? On the whole body level, small or non-significant differences in heat balance between young and old are reported (Stapleton et al., 2014). Importantly, the Stapleton study was conducted at 40 degrees C and 15%rh at metabolic rates ranging from 300 to 500W. Your results (lines 166-169) seem to suggest that this should be impossible for older adults (65 yoa in Stapelton et al., 2014).

Response: The results on Line 166-169 in the original manuscript were referencing liveability, which is about sustaining a certain rate of physical activity over a long period without seeing a sustained positive rate of heat storage. So we are not saying the activities are impossible, just that they are not possible without having a continuously rising T_{core}. So indeed, under very hot and dry conditions (45°C and 10% RH), an older adult would not be able to do more physical activity than 1.5 METs without a sustained positive rate of heat storage. This does not mean they cannot do any physical activity (they can do bouts of activity), but they should not do it for an extended period.

Related to the question of why age impacts E_{max}, this relates to the fact that the peripheral sensitivity to acetylcholine, which is responsible for sweating, is reduced with primary aging on average across the body. This reduces the overall ω_{max} , which reduces E_{max,wet}. Thank you for bringing up the Stapleton study. It seems that there was not much difference in the results between middle-aged (mean age 49)

and older males (mean age 65), but there was a difference with the young males (mean age 21). So, those findings support our paper here. Further, the Stapleton study is on people exercising in intermittent bouts (300W, 400W, 500W, as mentioned in the comment). Our survivability analysis looks at full rest (~100W), so we don't think the Stapleton results are comparable for survivability. However, referring to liveability, our analysis isn't too far off that of the Stapleton paper, where at the same RH and T_{air} , (40C and 15%), we say M_{max} is ~2.3–3.0METs. In the Stapleton paper, the lowest intensity was ~300W or 3.5Mets. Thus, we feel it is a decent agreement. This study has also been cited as an example of physiologic comparisons in Table 3.

Also, to clarify, we aren't looking at progressive effects of age, just a categorical young and old group comparison, which is a significant advancement on the existing models used in survivability analyses.

5. Would use of the UTCI model be a better approach? It has recently been validated against empirical data. Please see: Bröde P, Kampmann B. Temperature-Humidity-Dependent Wind Effects on Physiological Heat Strain of Moderately Exercising Individuals Reproduced by the Universal Thermal Climate Index (UTCI). *Biology* (Basel). 2023 May 31;12(6):802.

Overall, it seems that a small change in any one of many model assumptions could significantly alter your results, interpretations, and conclusions.

Response: We do not believe the UTCI wouldn't be better. Indeed, the model may be more sensitive to certain changes than others due to how different humans respond to heat. However, some fundamental issues in not using it is that all UTCI estimations are for a young and healthy 73.5 kg man, with 1.86 m² of body surface area and 14% body fat working a metabolic rate of 3 METs. It is, therefore, unable to generate any heat stress risk assessments for women, anyone at rest (most likely activity in a survival scenario), or older adults (who are at most risk). We also cannot use it for the liveability analysis because with UTCI we can't back calculate the MET.

Being able to look at diverse populations is one of the main reasons we set out to do this paper and model different types of people. However, we also recognize that certain assumptions, such as skin wettedness, may change our result as we change the value, yet such values can't even be modified in the UTCI. We have used the best values that we have from the literature and chamber studies and ensure that we remain conservative in our assumptions, similar to that of the studies applying the wet bulb of the 35°C approach. In recognizing this fact, numerous chamber experiments are going on in multiple countries working to address these specific values of skin wettedness, skin temperature, and sweat rate in different populations at different critical temperatures so that the types of modelling and approaches used (such as in this paper) here can be more accurate. The flexible approach we have developed will allow that in the future. Overall, we have used the values that 1) we are most confident in from the literature that is justifiable to use (with full citation support), and 2) tried to use assumptions (currently) that allow us to compare with the T_w of 35°C survivability model, with the full future model capability of varying these by person and in space or time with the approach we have created. Others will be able to do the same thing with the model, and the model can be even more complex for additional types of people once more human subjects' chamber data are available. It is a clear example of interdisciplinary research coming together to drive new questions and capabilities.

REVIEWERS' COMMENTS

Reviewer #1 (Remarks to the Author):

The authors have sufficiently addressed my concerns, in what to my mind is an excellent revision. As a minor quibble, I feel they have overstated the degree to which the literature has tended to claim 35C wet bulb is the be all and end all survivability threshold. I also felt that my initial critique of their presentation of their model results as lacking sufficient caveats led in their rebuttal to an assumption that I thought 35C wet bulb (or relying on wet bulb only in general) was a superior approach. That certainly is not my view--I was just hoping to bring out more caveats about their approach, something the authors have now added. My sense is that reviewer three may at the end of their comments have been on to something that could use a little more caveating, but as that material (human physiology and heat shedding) is far outside my areas of expertise, I defer to Reviewer 3.

Reviewer #2 (Remarks to the Author):

The authors have successfully responded to each of the reviewer's comments resulting in a significant improvement of the manuscript. Therefore, I recommend the manuscript for publication.

Reviewer #3 (Remarks to the Author):

I appreciate the efforts at revision and responses of the authors. I have no further questions or comments.

Response to Reviewers:

Reviewer #1 (Remarks to the Author):

The authors have sufficiently addressed my concerns, in what to my mind is an excellent revision. As a minor quibble, I feel they have overstated the degree to which the literature has tended to claim 35C wet bulb is the be all and end all survivability threshold. I also felt that my initial critique of their presentation of their model results as lacking sufficient caveats led in their rebuttal to an assumption that I thought 35C wet bulb (or relying on wet bulb only in general) was a superior approach. That certainly is not my view--I was just hoping to bring out more caveats about their approach, something the authors have now added. My sense is that reviewer three may at the end of their comments have been on to something that could use a little more caveating, but as that material (human physiology and heat shedding) is far outside my areas of expertise, I defer to Reviewer 3.

Response:

Thank you for this helpful feedback. It seems we may have overdone our comments on trying to create clarity for the reviewer regarding the use and applicability of the wet-bulb, and we have stated the caveats now too much. To address this comment, we have reviewed the entire paper and areas where the caveats and nuances are discussed and reduced the amount to which we state these concerns (reducing repetition) to balance things out more. Now we feel we have reached somewhere in between our initial submission 1 and submission 2, thus finding a happy medium on this topic. This is now reflected in the final article and has allowed us to also shorten the manuscript slightly.

Reviewer #2 (Remarks to the Author):

The authors have successfully responded to each of the reviewer's comments resulting in a significant improvement of the manuscript. Therefore, I recommend the manuscript for publication.

Response:

Thank you. We are glad the reviewer is happy with the improvements.

Reviewer #3 (Remarks to the Author):

I appreciate the efforts at revision and responses of the authors. I have no further questions or comments.

Response:

Thank you. We are glad the reviewer is happy with the improvements.